# GRADIENT-SIGN MASKING FOR TASK VECTOR TRANSPORT ACROSS PRE-TRAINED MODELS

**Filippo Rinaldi**[1]**, Aniello Panariello**[1]**, Giacomo Salici**[1]**, Fengyuan Liu**[2,3]**,**
**Marco Ciccone**[2]**, Angelo Porrello**[1]**, Simone Calderara**[1]

[1]University of Modena and Reggio Emilia, Italy
[2]Vector Institute, Canada      [3]University of Toronto, Canada

`name.surname@unimore.it, name.surname@vectorinstitute.ai`

## ABSTRACT

When a new release of a foundation model is published, practitioners typically need to repeat fine-tuning, even if the same task was already tackled in the previous version. A promising alternative is to reuse the parameter changes (*i.e.*, task vectors) that capture how a model adapts to a specific task. However, these vectors often fail to transfer across different pre-trained models because their parameter spaces are misaligned. In this work, we show that successful transfer depends strongly on the gradient-sign structure of the new model. Based on this insight, we propose GradFix, which approximates the ideal sign structure and leverages it to transfer knowledge using only a handful of labeled samples. Notably, this requires no additional fine-tuning: we only compute a few target-model gradients without parameter updates and mask the source task vector accordingly. This yields an update that is locally aligned with the target loss landscape, effectively rebasing the task vector onto the new pre-training. We provide a theoretical guarantee that our method ensures first-order descent. Empirically, we demonstrate significant performance gains on vision and language benchmarks, consistently outperforming naive task vector addition and few-shot fine-tuning. We further show that transporting task vectors improves multi-task and multi-source model merging. Code is available at `https://github.com/fillo-rinaldi/GradFix`.

## 1 INTRODUCTION

Over the past few years, the paradigm in deep learning has shifted from training models from scratch to fine-tuning large pre-trained models. Adapting these large models to downstream tasks is advantageous, as it leads to stronger performance at a fraction of the cost. Such a shift has been evident in natural language processing and computer vision, where pre-trained models such as BERT (Devlin et al., 2019), CLIP (Radford et al., 2021), and their successors (OpenAI et al., 2023; Liu et al., 2023) have become the standard starting point for developing new applications.

Since companies and researchers often update checkpoints using more data or improved training pipelines, practitioners frequently need to repeat fine-tuning on the same downstream tasks. This creates redundancy: the work invested in adapting one release is not directly reusable on the next. To address this issue, several lines of research have investigated how to systematically relate or transfer knowledge across parameter spaces. The model rebasin literature (Ainsworth et al., 2023; Rinaldi et al., 2025) studies how to align and merge independently trained models by exploiting permutation symmetries in their parameters. In parallel, Task Arithmetic (Ilharco et al., 2023; Ortiz-Jiménez et al., 2023; Yadav et al., 2023; Panariello et al., 2025b; Porrello et al., 2026) has shown that task vectors (*i.e.*, the difference $\tau = \theta^{ft} - \theta^0$ between base and fine-tuned parameters) can be added, subtracted, and merged across models to induce new capabilities. On a similar note, the literature on *mode connectivity* (Garipov et al., 2018; Frankle et al., 2020) demonstrates that different fine-tuned solutions can be linked by low-loss paths, highlighting that model parameters encode highly structured and transferable representations. Together, these advances suggest that parameters encode *rich* and *transferable* structure that can be systematically manipulated to obtain the desired behavior at reduced cost.

In particular, Rinaldi et al. (2025) formalizes this setting and proposes a technique to transport task vectors across transformer-based architectures. Yet a large gap remains between the transported fine-tune and an actual fine-tuned model, which highlights a key challenge: *while task vectors are informative about adaptation, their direct transfer across different pre-trained models is not guaranteed to align with the loss geometry of the target model*. In fact, naive transfer may introduce harmful directions in parameter space, *i.e.*, components of the task vector that are misaligned with the descent directions of the target loss, thus increasing the loss and limiting its effectiveness. Addressing this problem is crucial both for reducing the cost of adapting rapidly evolving foundation models and for enabling their use in low-data regimes, where re-running full fine-tuning is infeasible.

In this work, we introduce a framework for transporting task-specific knowledge across pre-trained models using **gradient-sign masking**. Our key insight is that, although a fine-tuning trajectory encodes valuable task information, its effectiveness on a new pre-trained model depends on the local loss geometry of the target. Inspired by findings from the optimization and distributed training literature (Bernstein et al., 2018; Alistarh et al., 2017), we exploit the observation that the sign of the gradient provides a robust surrogate for the descent direction. Leveraging this insight, we introduce a simple yet effective method to transport a task vector from a source model to a target pre-trained model: we mask the source task vector using the gradient signs of the target, keeping only the components aligned with the target's local loss landscape. We further provide a formal guarantee that, to first order, this transported update reduces the target loss, ensuring a principled safeguard against harmful or misaligned transfer.

Empirically, we show that this method enables highly effective transfer of fine-tuning knowledge from an outdated pre-trained model to a newer one, even in the low-data regime where gradients can only be estimated from a handful of samples, partially closing the gap between naive transfer and full fine-tuning on the target model. We also evaluate GradFix in model-merging pipelines, showing gains in both multi-task and multi-source transport settings. Our contributions are:

- We establish a theoretical connection between the *oracle task vector*, the ideal fine-tuning update on the target model, and quantities we can actually compute, namely the source task vector and the gradient at the zero-shot target model. We show that the sign of the zero-shot gradient provides a reliable proxy for the descent directions encoded in the target model.

- Building on this insight, we propose **GradFix**, a simple mechanism that filters the source task vector using the target model's local loss geometry. We formally prove that, to first order, the transported update reduces the target loss.

- We empirically show that our method enables effective transport of fine-tuning knowledge across pre-trained models in both vision and text domains, even in the *low-data regime* where gradients must be estimated from only a handful of samples. We further validate that GradFix improves model-merging performance in both multi-task and multi-source settings, showing that the transported updates remain useful beyond single-task transfer.

## 2  RELATED WORK

**Model merging.** Prior work studies how to merge fine-tuned checkpoints from the same pre-trained model. Model soups show that simple weight averaging can improve generalization (Wortsman et al., 2022). Task Arithmetic interprets fine-tuning deltas as task vectors that can be combined through linear operations (Ilharco et al., 2023), and TIES-Merging addresses conflicts by enforcing sign consistency (Yadav et al., 2023). Recent extensions include curvature-aware composition for improved adaptation (Porrello et al., 2025) and modular embedding recomposition for more flexible transfer across diverse tasks (Panariello et al., 2025a).

**Model rebasin.** A different family of methods focuses on explicit rebasing, aligning independently pre-trained models into a shared parameterization so that task vectors can be transported across different pre-trainings. Git Re-Basin introduced permutation matching to map two networks into a common basin (Ainsworth et al., 2023). For transformers, Imfeld et al. (2024) applies Optimal Transport to softly align components, while Rinaldi et al. (2025) proposes permutation- and spectral-based procedures that enable task vector transfer across distinct pre-trained models. Extensions such as permutation least-squares (Nasery et al., 2025) and supernet formulations (Stoica et al., 2024) address more heterogeneous or large-scale settings.

**Gradient information.** A complementary line of work studies the utility of gradient signs and compressed gradient information. SignSGD (and its majority-vote variant) shows that one-bit sign information can suffice for convergence in distributed settings (Bernstein et al., 2018), while quantization and error-feedback analyses formalize guarantees for compressed updates (Alistarh et al., 2017; Karimireddy et al., 2019). More recent methods leverage gradient magnitudes or sign statistics for efficient adaptation: Gradient-Mask Tuning masks low-importance parameters during LLM fine-tuning (Li et al., 2025), and sign-based federated variants weight client updates to address heterogeneity (Park et al., 2024).

## 3 PRELIMINARIES

Let $\theta_A$ and $\theta_B$ denote the parameters of the same architecture, pre-trained on different datasets (or with different hyperparameters). The fine-tuned $\theta_A$ on a downstream task is denoted by $\theta_A^{ft}$.

**Model rebasin.** The goal of rebasin (Ainsworth et al., 2023) is to align two independently trained models by mapping the parameters of one into the loss basin of the other, so that they become functionally compatible. This setting concerns only the pre-trained weights, without involving fine-tuning updates.

**Task Arithmetic.** A complementary perspective to model rebasin is offered by *Task Arithmetic* (Ilharco et al., 2023; Yadav et al., 2023), which studies linear operations on task-specific parameter updates. Given a pre-trained model $\theta^0$ and a fine-tuned counterpart $\theta^\star$, the difference vector in parameter space $\tau := \theta^\star - \theta^0$ is called a *task vector*. Task vectors describe how a base model adapts to the task and can be added, subtracted, or merged to induce new behaviors. This setting usually assumes that all models share the same initialization $\theta^0$, which ensures comparability across tasks.

**Our setting.** In contrast, our goal is to apply task-vector transfer when the target and source models do not share the same initialization: we want to transfer $\tau_A = \theta_A^{ft} - \theta_A$ from a source pre-trained model $\theta_A$ to a different pre-trained model $\theta_B$. This connects to rebasin in that the bases differ, but unlike traditional rebasin approaches, we do not seek to explicitly find a permutation to align parameters. Instead, we ask:

*Which components of $\tau_A$ are truly transferable, and which would instead harm $\theta_B$?*

This question motivates our method, which leverages the local gradients of $\theta_B$ to selectively filter $\tau_A$ into a compatible and transferable update, effectively performing direct task vector transportation.

## 4 METHOD

Here we introduce GradFix , a framework for transferring task vectors across different pre-trained models by filtering them with gradient information from the target model. As a conceptual starting point, we first consider an *oracle* setting where the target task vector obtained from full fine-tuning defines the ideal transferable directions (Sec. 4.1). We then approximate this oracle with a *single gradient step* on the target model, using anti-gradient signs to estimate the direction of the full fine-tuning trajectory. This yields a *gradient-sign mask* that selectively filters the source task vector into a compatible update (Sec. 4.2). Finally, we extend the approach to the limited-data regime, where gradients are estimated from only a handful of labeled samples (Sec. 4.3).

### 4.1 GRADFIX (GRADIENT-SIGN MASKING)

We begin by considering an ideal scenario where the true fine-tuned task vector $\tau_B = \theta_B^{ft} - \theta_B$ of model $B$ and the full target dataset $\mathcal{D}$ are available. This vector represents the optimal parameter change to adapt $B$ to $\mathcal{D}$. In this setting, we can construct a mask that retains only the components of a candidate update (*e.g.*, $\tau_A$) that are aligned with $\tau_B$, ensuring that every retained coordinate contributes to decreasing the loss. In other words, $\tau_B$ (or its sign structure) defines the gold standard for locally beneficial directions. Formally, we define the oracle mask $\boldsymbol{m}^\star \in \{0,1\}^d$, where $d$ is the total number of model parameters and $i \in \{1, \ldots, d\}$ indexes a parameter coordinate:

$$m_i^\star = \mathbb{1}\{\text{sign}(\tau_{A,i}) = \text{sign}(\tau_{B,i})\}. \tag{1}$$

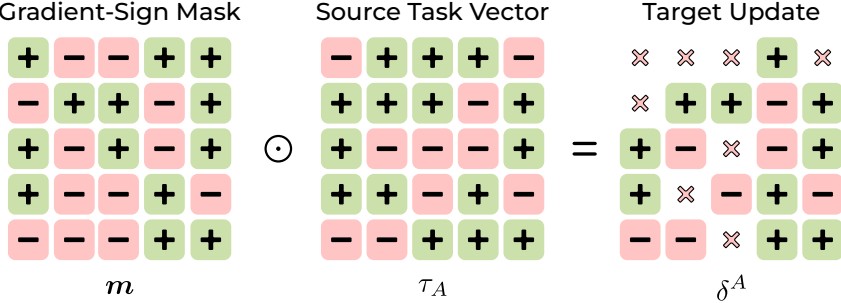

Figure 1: Illustration of our masking procedure. The gradient mask $\boldsymbol{m}$ suppresses harmful directions in the task vector $\tau_A$ while preserving those aligned with the target model.

As shown in Fig. 1, applying this mask to $\tau_A$ produces the oracle-masked update $\delta^\star$, which preserves only the components consistent with $\tau_B$:

$$\delta^\star := \boldsymbol{m}^\star \odot \tau_A, \tag{2}$$

where $\odot$ denotes element-wise multiplication. This vector $\delta^\star$ represents a reliable transfer of $\tau_A$ onto $\theta_B$, since it filters out all components of $\tau_A$ that are misaligned with the true adaptation directions of $B$. In practice, however, $\tau_B$ (and thus $\delta^\star$) is unavailable because it requires access to the fine-tuned target model $\theta_B^{ft}$, which defeats the purpose of transporting the solution from $A$ to $B$. To approximate this ideal mask, we use the gradient of the zero-shot target model as a surrogate for $\tau_B$, since it captures locally beneficial directions:

$$\boldsymbol{g} := \nabla_\theta \mathcal{L}(\theta_B), \quad \mathcal{L}(\theta) := \mathbb{E}_{(x,y)\sim\mathcal{D}}[\ell(f_\theta(x), y)], \tag{3}$$

where $\ell$ is the training objective (*e.g.*, cross-entropy) and $(x, y)$ is a labeled example from $\mathcal{D}$. Based on this gradient, we define the gradient-sign mask $\boldsymbol{m}$, which retains only the components of $\tau_A$ whose sign matches that of the corresponding anti-gradient coordinate:

$$m_i := \mathbb{1}\{\mathrm{sign}(\tau_{A,i}) = \mathrm{sign}(-g_i)\}. \tag{4}$$

Intuitively, $-\boldsymbol{g}$ acts as a signal for local alignment with the loss geometry of $B$. Notably, in the idealized setting where $B$ is fine-tuned using full-batch gradient descent for a single epoch, the resulting task vector $\tau_B$ is proportional to $-\boldsymbol{g}$, so the gradient-sign mask coincides with the oracle. This observation justifies using the gradient-sign mask as an approximation of the ideal update, even when only a few labeled examples are available. The mask retains only components of $\tau_A$ whose sign matches the anti-gradient of $\mathcal{L}(\theta_B)$, pruning coordinates that would increase the loss for target model $B$. In this way, the gradient-sign mask serves as a practical surrogate for the trajectory-informed directions encoded in the unavailable $\tau_B$.

## 4.2 Transporting the Update

Given the gradient-sign mask $\boldsymbol{m}$ from Eq. (4), we define the updated target parameters by directly applying the masked task vector with a scaling factor $\alpha > 0$:

$$\theta_B^{\mathrm{trans}} = \theta_B + \delta^A, \quad \delta^A := \alpha\,(\boldsymbol{m} \odot \tau_A). \tag{5}$$

We explicitly use the convention $\boldsymbol{g} = \nabla_\theta \mathcal{L}(\theta_B)$ (ascent direction), so useful transport directions should align with $-\boldsymbol{g}$. It is important to note that $\tau_A$ points in a descent direction for model $A$, whereas the gradient $\boldsymbol{g}$ of the target model points in the ascent direction of its loss. By selecting only coordinates aligned with $-\boldsymbol{g}$ and then adding $\delta^A$, each retained component moves along a descent-aligned direction for $B$. In contrast, $\delta^\star$ is oracle-aligned with the target task direction because its mask is constructed from $\tau_B$.

This construction induces a coordinate-wise filtering effect: we do not alter the direction of retained entries, but only suppress coordinates that are sign-incompatible with the target anti-gradient. As a result, the transported update preserves task information from $\tau_A$ while reducing the risk of injecting locally harmful directions into $\theta_B$. We now formalize this intuition with a first-order loss analysis.

**Descent guarantee.** To understand why this gradient masking provides effective transfer, we analyze its effect on the loss of the target model $B$. Consider the transported update from Eq. (5). By expanding the target loss $\mathcal{L}$ around $\theta_B$ via a first-order Taylor approximation, we obtain:

$$\mathcal{L}(\theta_B + \delta^A) \approx \mathcal{L}(\theta_B) + \boldsymbol{g}^\top \delta^A, \quad \text{where } \boldsymbol{g} = \nabla_\theta \mathcal{L}(\theta_B). \tag{6}$$

The sign of the inner product $\boldsymbol{g}^\top \delta^A$ determines whether the update increases or decreases the loss to first order. By construction, the gradient-sign mask $\boldsymbol{m}$ retains only components of $\tau_A$ that are aligned with $-\boldsymbol{g}$. Concretely, for each coordinate $i$, we have:

$$g_i \cdot (m_i \tau_{A,i}) = \begin{cases} -|g_i|\,|\tau_{A,i}|, & \text{if } \operatorname{sign}(\tau_{A,i}) = \operatorname{sign}(-g_i), \\ 0, & \text{otherwise,} \end{cases} \tag{7}$$

which is always nonpositive. Coordinates with $g_i = 0$ or $\tau_{A,i} = 0$ are naturally covered by this expression and contribute zero. Therefore, the overall inner product satisfies the following:

$$\boldsymbol{g}^\top \delta^A = -\alpha \sum_i m_i |g_i|\,|\tau_{A,i}| \;\le\; 0. \tag{8}$$

Thus, for sufficiently small $\alpha$, the update $\delta^A$ is guaranteed to be a descent direction for $\mathcal{L}$. Practically, the mask removes all sign-mismatched components of $\tau_A$, so that every retained entry contributes to reducing the loss. Without masking, $\tau_A$ could contain harmful directions that increase the loss for $B$; with masking, the transported update is locally aligned with the descent geometry of the target model.

### 4.3 LIMITED DATA REGIME

In Sec. 4.1, we have assumed access to the full target dataset $\mathcal{D}$ to compute the gradient $\boldsymbol{g}$ at the zero-shot target model $\theta_B$. In practice, one of the main motivations for task vector transport is the *few-shot* or limited data regime. If we had access to the entire dataset, we could directly fine-tune $\theta_B$ to obtain $\theta_B^{ft}$, making task vector transfer unnecessary in that setting.

When only a small number of samples is available, we estimate the anti-gradient signs using a subset of labeled examples. Let $\mathcal{D}_s \subset \mathcal{D}$ denote a small subset of $N$ samples. For each parameter coordinate $i$, we compute the sign of the anti-gradient via **majority voting** across these samples:

$$\hat{s}_i = \operatorname{sign}\left( -\sum_{(x_n, y_n) \in \mathcal{D}_s} \operatorname{sign}\left( \nabla_\theta \ell\left( f_{\theta_B}(x_n), y_n \right) \right) \right). \tag{9}$$

**Lemma** (Concentration of Majority Vote Sign Estimator). *Let $p_i = \Pr[\operatorname{sign}(\nabla_\theta \ell(f_{\theta_B}(x), y)) = \operatorname{sign}(g_i)]$ denote the probability that a single-sample gradient sign matches the true gradient sign at coordinate $i$. Then, under mild independence assumptions and for $p_i > 1/2$, the majority-vote estimator satisfies:*

$$\Pr\left[ \hat{s}_i = \operatorname{sign}(-g_i) \right] \ge 1 - \exp\left( -2N(p_i - 1/2)^2 \right), \tag{10}$$

*which shows that the estimated sign concentrates around the true anti-gradient direction as the number of samples $N$ grows.*

We provide a proof of this lemma in Appendix A, which uses Hoeffding's inequality (Hoeffding, 1963). In practice, even a few samples provide a robust estimate of the true descent direction. Each gradient acts as a vote for the correct sign, and majority voting filters out noisy or conflicting directions. This implies that, with high probability, the masked task vector $\delta^A$ points in a descent direction, preserving the first-order loss reduction behavior of the full-data update. As shown in Sec. 5.3, this approach is robust to small sample sizes, making it particularly attractive when direct fine-tuning of $\theta_B$ is expensive or prone to overfitting. Compared to mean-based aggregation, majority voting is less sensitive to outlier magnitudes because it depends only on sign frequency and provides a more stable transfer (Sec. 5.4).

Algorithm 1 outlines the gradient-sign masked task vector transport procedure, showing how the source task vector is selectively applied to the target model using only a small subset of labeled data.

---

**Algorithm 1** Gradient-Sign Masked Transport

---

**Require:** Source model $\theta_A$, $\theta_A^{ft}$, target model $\theta_B$, target data subset $\mathcal{D}_s$, scaling $\alpha$.
**Ensure:** Transported model $\theta_B^{trans}$

1: Compute source task vector: $\tau_A \leftarrow \theta_A^{ft} - \theta_A$
2: **for** $(x_n, y_n) \in \mathcal{D}_s$ **do**
3: $\quad g^{(n)} \leftarrow \nabla_\theta \ell(f_{\theta_B}(x_n), y_n)$
4: Compute anti-gradient signs $\hat{s}_i$ by majority voting $\hfill \triangleright$ Eq. (9)
5: Build gradient-sign mask: $m_i \leftarrow \mathbb{1}\{\text{sign}(\tau_{A,i}) = \hat{s}_i\}$
6: Compute transported update: $\delta^A \leftarrow \alpha\,(\boldsymbol{m} \odot \tau_A)$
7: **return** $\theta_B^{trans} \leftarrow \theta_B + \delta^A$ $\hfill \triangleright$ Updated target model

---

## 5 Experimental Results

**Implementation details.** For the vision settings, we consider CLIP ViT-B/16 and ViT-L/14 Vision Transformers (Radford et al., 2021), implemented in Open-CLIP (Cherti et al., 2023). The source pre-trained weights are denoted $\theta_A$ and the target pre-trained weights are denoted $\theta_B$. For ViT-B/16, $\theta_A$ was pre-trained on Datacomp XL (`s13b`, `b90k`) and $\theta_B$ on LAION-2B (`s34b`, `b88k`). For ViT-L/14, $\theta_A$ was pre-trained on Datacomp XL (`s13b`, `b90k`) and $\theta_B$ on LAION-2B (`s32b`, `b82k`). For the language settings, we use T5-base variants (Raffel et al., 2020). As $\theta_A$, we use `T5v1.1`, pre-trained on C4 (Raffel et al., 2020) without supervised training. As $\theta_B$, we use `FLAN-T5` (Chung et al., 2024), pre-trained and instruction-tuned on several datasets, including GSM8K (Cobbe et al., 2021), AQUA-RAT (Ling et al., 2017), and LAMBADA (Paperno et al., 2016). Task vectors were obtained following the fine-tuning protocol of Ilharco et al. (2023): 2000 iterations, batch size 128, learning rate $1 \times 10^{-5}$, cosine annealing with 200 warm-up steps, AdamW optimizer (Loshchilov & Hutter, 2019), weight decay 0.1. The text encoder backbone was kept frozen following Cherti et al. (2023). For fairness, all compared transport methods use the same source/target checkpoints and the same supervision budget $\mathcal{D}_s$, and results are aggregated across random seeds. The $\alpha$ is chosen through a validation set following standard model merging pipelines, more details in Appendix E.

**Baselines.** We evaluate our method against several baselines. As a lower bound, we consider the zero-shot target model ($\theta_B$ *zero-shot*), *i.e.*, the base model without fine-tuning. As an upper bound, we report $\theta_B + \delta^\star$, obtained by adding the source task vector $\tau_A$ masked with the signs of the true task vector $\tau_B$ to the target model. We also include the performance of the fully fine-tuned target model ($\theta_B$ *fine-tune*) and the naive Task Arithmetic transport ($\theta_B + \tau_A$). In addition, we compare against *TransFusion* (Rinaldi et al., 2025), which transports task vectors across transformer-based models via permutation alignment. Finally, we report the performance of a target model fine-tuned with the same number of randomly sampled examples per class $|\mathcal{D}_s^c|$ used by our approach.

**Supervision budget $\mathcal{D}_s$.** In all experiments, the subset $\mathcal{D}_s$ is drawn from the full downstream fine-tuning dataset $\mathcal{D}$ and constitutes only a fraction of its size. Throughout the tables, $|\mathcal{D}_s^c|$ indicates the number of examples *per class* used to estimate anti-gradient signs for the target model $\theta_B$. The corresponding proportions of $\mathcal{D}$ used to form $\mathcal{D}_s$ are provided in Appendix D. Additionally, we include an ablation study in Appendix C.2 analyzing the impact of different subset selection heuristics on anti-gradient sign estimation.

### 5.1 Transport Experiments

**Transport in the vision setting.** Table 1 summarizes the results of task vector transport across CLIP ViT-B/16 and CLIP ViT-L/14 architectures, averaged over multiple random seeds that determine the composition of the sampled $\mathcal{D}_s$ (standard deviations are reported in Appendix B.1). Our GradFix, denoted by $\theta_B + \delta^A$, yields a consistent improvement over naive task vector addition ($\theta_B + \tau_A$) even when using a single sample per class to approximate true anti-gradient signs. Notably, the naive addition performs nearly at the level of zero-shot initialization and fails to transfer meaningful task knowledge. This confirms that GradFix effectively suppresses misaligned components of $\tau_A$, preventing negative transfer due to pre-training mismatch.

Table 1: Cross-dataset performance comparison for ViT-B/16 and ViT-L/14 models.

| Model | $|\mathcal{D}_s^c|$ | EUROSAT | | SVHN | | GTSRB | | RESISC45 | | DTD | |
| | | B/16 | L/14 | B/16 | L/14 | B/16 | L/14 | B/16 | L/14 | B/16 | L/14 |
|---|---|---|---|---|---|---|---|---|---|---|---|
| $\theta_B$ *zero-shot* | - | 49.41 | 62.80 | 50.58 | 37.28 | 48.29 | 56.12 | 67.98 | 73.12 | 55.96 | 63.35 |
| $\theta_B$ *fine-tune* | - | 98.70 | 98.95 | 97.45 | 97.80 | 98.65 | 99.16 | 95.66 | 97.06 | 83.19 | 83.56 |
| $\theta_B + \tau_A$ | - | 49.58 | 62.77 | 50.84 | 39.09 | 49.31 | 56.03 | 67.87 | 73.49 | 56.27 | 63.56 |
| $\theta_B + \delta^\star$ | - | 95.06 | 96.75 | 92.04 | 92.60 | 82.92 | 88.65 | 87.06 | 90.30 | 71.44 | 72.66 |
| TransFusion | - | 50.12 | 63.21 | 53.26 | 37.38 | 50.24 | 56.78 | 67.99 | 73.36 | 56.70 | 64.10 |
| $\theta_B^{opt}$ | 1 | 56.61 | 64.65 | 61.32 | 62.51 | 56.08 | 63.97 | 69.25 | 74.54 | 56.21 | 63.76 |
| $\theta_B + \delta^A$ | 1 | 61.94 | 69.67 | 71.07 | 70.15 | 60.88 | 66.82 | 70.05 | 76.45 | 58.32 | 65.50 |
| $\theta_B^{opt}$ | 2 | 59.49 | 70.76 | 62.01 | 45.23 | 61.70 | 69.91 | 71.20 | 76.62 | 57.00 | 64.97 |
| $\theta_B + \delta^A$ | 2 | 65.07 | 74.10 | 70.19 | 54.31 | 64.33 | 71.55 | 71.42 | 76.97 | 58.51 | 66.10 |
| $\theta_B^{opt}$ | 5 | 61.99 | 69.75 | 67.03 | 67.11 | 63.08 | 73.25 | 73.01 | 75.41 | 59.65 | 66.72 |
| $\theta_B + \delta^A$ | 5 | 66.05 | 75.59 | 73.59 | 74.41 | 66.61 | 73.14 | 71.57 | 76.82 | 60.02 | 66.95 |

Table 2: Cross-dataset performance of T5 models on different NLP tasks.

| Model | $|\mathcal{D}_s^c|$ | SNLI | MNLI | RTE | QNLI | SCITAIL | AVG |
|---|---|---|---|---|---|---|---|
| $\theta_B$ *zero-shot* | - | 34.24 | 35.21 | 47.20 | 50.54 | 50.38 | 43.51 |
| $\theta_B$ *fine-tune* | - | 88.20 | 86.30 | 84.40 | 92.79 | 95.32 | 89.40 |
| $\theta_B + \tau_A$ | - | 31.61 | 30.75 | 47.36 | 50.52 | 50.46 | 42.12 |
| $\theta_B + \delta^\star$ | - | 58.69 | 69.97 | 72.93 | 65.32 | 62.38 | 65.86 |
| $\theta_B^{opt}$ | 50 | 35.09 | 26.05 | 47.29 | 51.45 | 51.78 | 42.33 |
| $\theta_B + \delta^A$ | 50 | 68.06 | 49.68 | 54.25 | 60.50 | 59.89 | 58.48 |

To further evaluate our approach, we compare it against few-shot fine-tuning of $\theta_B$, denoted as $\theta_B^{opt}$, using the same limited target samples and number of training steps (see Appendix F for a detailed computational cost analysis). This comparison isolates the effect of update construction: both methods observe the same supervision budget, but $\theta_B^{opt}$ relies on iterative parameter optimization, whereas GradFix applies a single masked transport step. GradFix achieves better performance on both ViT-B/16 and ViT-L/14, while exhibiting smaller variance across seeds with respect to few-shot fine-tuning (Tabs. 5 and 6). Moreover, as the $\mathcal{D}_s$ size increases, our method continues to provide stable gains, whereas $\theta_B^{opt}$ shows larger fluctuations due to subset composition. These trends suggest that GradFix is not only stronger on average, but also more reliable under realistic data-sampling variability. In practical low-shot settings, this matters because adaptation quality should remain stable even when the selected subset is not carefully curated. Overall, gradient-sign masking yields a more data-efficient and predictable adaptation mechanism, where robustness is achieved with a single forward-backward pass to obtain the mask $m$.

**Transport in the language setting.** Table 2 reports results on task vector transport across T5 models evaluated on closed-vocabulary text classification benchmarks. While direct addition of $\tau_A$ to $\theta_B$ fails to transfer knowledge effectively, our method substantially closes the gap toward full fine-tuning, confirming its ability to identify and retain task-relevant directions. Notably, the relative improvement over naive transfer is even larger than in the vision setting, suggesting that sign-based filtering is particularly beneficial when source and target pre-training objectives differ more strongly. This is a challenging regime, since transferring updates between T5v1.1 and FLAN-T5 combines both pre-training and instruction-tuning mismatch. Despite this shift, GradFix remains effective with the same lightweight procedure used in vision. These results confirm that the method is architecture-agnostic in practice and supports reliable task-vector transport in the language domain as well.

Table 3: Merging experiments on ViT-B/16 in multi-task and multi-source settings.

| Pipeline | EUROSAT | SVHN | GTSRB | RESISC45 | DTD | AVG |
|---|---|---|---|---|---|---|
| $\theta_B$ *zero-shot* | 49.41 | 50.58 | 48.29 | 67.98 | 55.96 | 54.44 |
| **Multi-task**  $(\{\tau_{A,j}\}_{j=1}^T \rightarrow \theta_B)$ | | | | | | |
| $\theta_B + \tau_{A,j}$ | 49.58 | 50.84 | 49.31 | 67.87 | 56.27 | 54.77 |
| $\theta_B + \delta_j^A$ | 65.07 | 70.19 | **64.33** | 71.42 | **58.51** | 65.90 |
| *Task Arithmetic* | | | | | | |
|   Baseline | 49.31 | 50.99 | 48.73 | 68.05 | 56.54 | 54.73 |
|   Mask-then-Merge | 55.90 | 71.56 | 59.65 | 71.40 | 57.66 | 63.23 |
|   Merge-then-Mask | 65.37 | 72.10 | 59.55 | 71.16 | 57.07 | 65.05 |
| *TIES-Merging* | | | | | | |
|   Baseline | 49.15 | 50.75 | 48.95 | 67.97 | 56.54 | 54.67 |
|   Mask-then-Merge | 50.41 | 64.28 | 54.05 | 69.51 | 57.13 | 59.08 |
|   Merge-then-Mask | **65.62** | **72.42** | 62.73 | **71.57** | 57.77 | **66.02** |
| **Multi-source**  $(\{\tau_{A_k}\}_{k=1}^K \rightarrow \theta_B)$ | | | | | | |
| *Task Arithmetic* | | | | | | |
|   Baseline | 36.94 | 35.09 | 30.77 | 50.54 | 44.73 | 39.61 |
|   Merge-then-Mask | 12.52 | 15.94 | 4.20 | 3.97 | 2.93 | 7.91 |
|   Mask-then-Merge | **65.96** | **72.97** | **65.80** | 71.30 | 61.01 | **67.41** |
| *TIES-Merging* | | | | | | |
|   Baseline | 12.69 | 10.39 | 2.85 | 5.81 | 16.33 | 9.61 |
|   Merge-then-Mask | 14.46 | 15.94 | 3.09 | 3.35 | 2.66 | 7.90 |
|   Mask-then-Merge | 65.17 | 72.58 | 65.36 | **71.40** | **61.12** | 67.13 |

## 5.2 TASK VECTOR TRANSPORT FOR MODEL MERGING

We evaluate GradFix in combination with model-merging methods, specifically Task Arithmetic (Ilharco et al., 2023) and TIES-Merging (Yadav et al., 2023). We consider two settings: *multi-task* (one source model, multiple tasks) and *multi-source* (multiple source models, one task).

**Multi-task experiments.** For this setting, all task vectors are extracted from the same source pre-trained model $\theta_A$ (same pre-training, different downstream tasks) and transported to a fixed target pre-trained model $\theta_B$. For task vectors from distinct tasks, we compare two pipelines. **Mask–then–Merge**: transport each task vector with GradFix and then merge. **Merge–then–Mask**: first merge task vectors into $\tau_{\text{merged}}$, then transport the merged vector using a consensus mask computed by estimating per-parameter anti-gradient signs on $\theta_B$ for each task and selecting the most frequent sign at each coordinate. We report results for this setting in Tab. 3. Here, $\theta_B + \tau_{A,j}$ and $\theta_B + \delta_j^A$ denote single-task references evaluated on task $j$, respectively using naive addition and GradFix. Direct Task Arithmetic and TIES merging without GradFix perform near zero-shot, indicating strong cross-model misalignment, while **Merge–then–Mask** gives the best results. Masking each vector first can discard coordinates that would complement each other after merging; merging first preserves them and lets GradFix align one coherent update with the target loss geometry.

**Multi-source experiments.** In the multi-source setting, we use $K = 5$ source models $\{\theta_{A_k}\}_{k=1}^K$ that are pre-trained on different data distributions and then fine-tuned on the same downstream task. We transport all resulting source task vectors to a single fixed target model $\theta_B$, whose pre-training remains unchanged during transport. Here, **Merge–then–Mask** is not expected to help: because all vectors correspond to the same task, the gradient-sign mask on $\theta_B$ is shared across sources, so consensus masking after merging is effectively equivalent to masking each source separately. We therefore use **Mask–then–Merge**: transport each source vector first, then merge the transported vectors. Table 3 shows that this recovers performance from the collapse of direct merging and also improves over single-source transport, suggesting that combining multiple transported updates provides a more robust descent direction.

Table 4: Performance of $\theta_B$ with oracle or estimated gradient signs under different mask strategies: **sign agreement** retains matching signs, **sign forcing** aligns all signs, **magnitude-scaled** uses the product of task and gradient magnitudes, and **random** assigns signs uniformly. Results averaged over seeds with $|\mathcal{D}_s^c| = 1$ on CLIP ViT-B/16. The table includes $\theta_B$ *zero-shot* and $\theta_B$ *fine-tune* as lower and upper reference bounds.

| Model | Mask Strategy | EUROSAT | RESISC45 | GTSRB | SVHN | DTD | AVG |
|---|---|---|---|---|---|---|---|
| $\theta_B$ *zero-shot* | - | 49.41 | 67.98 | 48.29 | 50.58 | 55.96 | 54.45 |
| $\theta_B$ *fine-tune* | - | 98.70 | 95.66 | 98.65 | 97.45 | 83.19 | 94.73 |
| $\theta_B + \delta^\star$ | sign agreement | 95.06 | 87.06 | 82.92 | 92.04 | 71.44 | 85.71 |
| | sign forcing | 97.95 | 93.51 | 95.94 | 96.60 | 80.59 | 92.92 |
| | magnitude-scaled | 49.92 | 67.94 | 51.63 | 50.78 | 56.01 | 55.25 |
| $\theta_B + \delta^A$ | sign agreement (ours) | 61.94 | 70.05 | 60.89 | 71.07 | 58.32 | 64.45 |
| | sign forcing | 61.32 | 70.10 | 60.91 | 70.52 | 58.05 | 64.18 |
| | magnitude-scaled | 49.51 | 68.06 | 49.20 | 50.71 | 56.03 | 54.70 |
| $\theta_B + \delta^A$ | random | 49.49 | 67.97 | 48.41 | 50.54 | 56.06 | 54.50 |

## 5.3 MASKING STRATEGIES

To analyze the effect of different mask construction strategies on the transport of the task vector $\tau_A$, we compare our primary masking method (sign agreement), with three alternatives: **sign forcing**, **magnitude-scaled**, and **random** masks. Table 4 reports results using 1 sample per class on ViT-B/16, averaged across multiple random seeds.

For both $\delta^A$ (Eq. (5)) and $\delta^\star$ (Eq. (2)), the mask $m$ determines which directions of $\tau_A$ are retained. Here, $i$ indexes parameter coordinates. In **sign agreement**, $m$ retains only the coordinates whose signs match those of the reference as in Eq. (4). In **sign forcing**, all signs of $\tau_A$ are aligned with the signs of the anti-gradient estimator $\hat{s}$, obtaining:

$$m_i^{sf} = \text{sign}(\tau_{A,i}) \cdot \hat{s}_i, \qquad \delta_i^{sf} = \alpha\left(m_i^{sf}\tau_{A,i}\right) = \alpha\left|\tau_{A,i}\right|\hat{s}_i. \qquad (11)$$

This flips entries that disagree with the reference and applies a forced-sign update. When the oracle $\tau_B$ is used as the reference, sign forcing generally outperforms sign agreement, as fully leveraging the true task direction maximizes transfer. In contrast, using few-shot anti-gradient-based estimates from $\theta_B$, sign agreement performs slightly better than sign forcing. This is consistent with the fact that the anti-gradient estimate is noisy; forcing all directions can propagate errors, while keeping only agreeing entries provides a more reliable mask. We also evaluated a **magnitude-scaled** strategy to determine whether leveraging magnitude information offers additional benefits. This strategy computes the mask based on the magnitude of both the source task vector and the target reference vector $\rho$ (where $\rho = \tau_B$ for the oracle and $\rho = -g$ for the estimate):

$$m_i^{ms} = \max(0, \tanh(\tau_{A,i} \cdot \rho_i)), \qquad \delta_i^{ms} = \alpha\left(m_i^{ms}\tau_{A,i}\right). \qquad (12)$$

This approach assigns a mask value near 1.0 only when components match in sign and possess significant magnitude, while suppressing mismatches. However, as shown in Tab. 4, this magnitude-aware strategy consistently underperforms sign agreement. Crucially, this failure persists even in the oracle setting ($\delta^\star$), which suggests that while *directions* are transferable across different pre-trainings, parameter *magnitudes* are highly specific to the local loss geometry of each basin. Consequently, enforcing magnitude consistency acts as an overly aggressive filter, discarding valid updates where the models agree on direction but differ on scale. In the limited-data regime ($\delta^A$), this issue is further exacerbated by the noise inherent in estimating gradient magnitudes from small datasets $\mathcal{D}_s$, confirming that the sign structure serves as the most robust proxy for alignment.

Finally, we evaluate **random mask**, where signs are sampled uniformly from $\mathcal{U}\{-1, +1\}$. Like magnitude-scaled masking, this approach performs close to the zero-shot baseline. This highlights that although sign-based masking outperforms magnitude-based filtering, the gain is not due to masking alone: random sign choices offer no useful signal. Thus, effective transfer relies strictly on the precise geometric alignment provided by the target anti-gradients, rather than just the sparsity of the update.

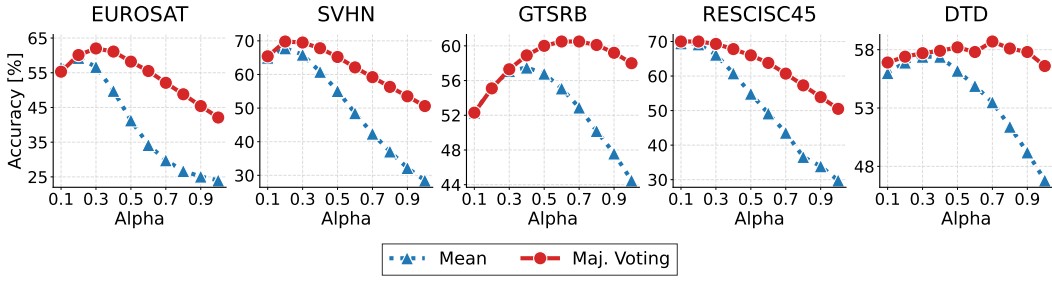

Figure 2: Accuracy across different $\alpha$ values for mean and majority-vote sign estimation.

## 5.4 Sensitivity to the Scaling Coefficient

We investigate the sensitivity of masked transport to the scaling factor $\alpha \in (0, 1]$, providing a proxy for how compatible and robust the transported task vector is with the target backbone. In addition to our proposed *majority voting* strategy, we consider a baseline where the estimated sign is taken as the sign of the averaged anti-gradient (*mean*). Results are reported in Fig. 2.

Across datasets, majority voting consistently outperforms the mean strategy for all values of $\alpha$, providing a more reliable approximation of the true gradient sign. Notably, majority voting yields smooth performance curves without sudden drops, and maintains higher accuracy over a broader range of $\alpha$. This difference arises from the aggregation mechanism; averaging gradients before thresholding is highly sensitive to variance and outliers, so even a small subset of misaligned samples can flip the estimated sign and destabilize updates as $\alpha$ grows. Majority voting, instead, depends only on the relative frequency of signs, which concentrates rapidly around the true direction with increasing samples (as shown in Appendix A). As a result, it is inherently more stable and preserves transfer accuracy even in few-shot or noisy regimes.

From a practical perspective, this robustness means that masked transport with majority voting does not require fine-grained tuning of $\alpha$ to achieve good performance. The method remains effective across a wide range of scaling choices, which is particularly valuable when adapting to new datasets where validation data or tuning budgets are limited.

## 5.5 Subset Data Selection

Beyond random sampling, we analyzed whether structured strategies for constructing $\mathcal{D}_s$ improve anti-gradient-sign estimation. We compared random selection with feature-based alternatives from CLIP embeddings: herding, $k$-medoids, and a coreset-style medoid-proximity greedy selection.

Across datasets, structured selectors yield small gains at very low budgets, while random sampling remains competitive and approaches their performance as the number of examples per class increases. Because random sampling adds no embedding/distance-computation overhead and does not require full target-data access, it is often preferable in constrained settings. Full details and curves are provided in Appendix C.2.

## 6 Conclusions & Future Work

In this work, we show that the sign structure of anti-gradients provides a powerful and reliable proxy for descent directions in the target loss landscape. The strong performance of our oracle-based analysis, which uses signs from the true task vector, validates this core insight and confirms that effective transfer is possible when the transported task vector is aligned with the target model's local geometry. **GradFix** approximates this oracle with only a handful of labeled samples to estimate anti-gradient signs, yielding large gains over naive transfer. While our approach is effective and robust in low-data regimes, the remaining gap to the oracle highlights clear directions for future work. In particular, future research can explore better anti-gradient sign estimators, stronger subset-selection strategies, and extensions to other architectures and transfer settings.

ACKNOWLEDGMENTS

This work was supported by the PNRR-M4C2 (PE00000013) project "FAIR - Future Artificial Intelligence Research". Angelo Porrello was partially supported by the Department of Engineering "Enzo Ferrari" through the FAR2025DIP program (CUP E93C25000370005). We acknowledge the CINECA award under the ISCRA initiative for access to high-performance computing resources and technical support.

REPRODUCIBILITY STATEMENT

We provide the codebase needed to reproduce our results. All hyperparameters used in our experiments are detailed in the appendix. We also provide complete proofs for all claims made in the paper, so that the results and statements can be independently verified.

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

USE OF LARGE LANGUAGE MODELS (LLMS)

We made limited use of a large language model only for light editorial refinement, such as adjusting phrasing, improving grammar and enhancing overall clarity. The model was not involved in generating research ideas, designing experiments, interpreting results, or authoring scientific content.

APPENDIX

# A    ANTI-GRADIENT SIGN ESTIMATOR GUARANTEE

We formalize why majority-vote estimation of anti-gradient signs provides a reliable proxy for true descent directions in the limited-data regime.

**Lemma** (Majority-Vote Gradient Sign). *Let $g_i := \nabla_{\theta_i} \mathcal{L}_B(\theta_B)$ denote the true target-loss gradient at parameter coordinate $i$, and let $g_i^{(n)}$ be i.i.d. per-sample gradients:*

$$g_i^{(n)} = g_i + \varepsilon_i^{(n)}, \quad \mathbb{E}[\varepsilon_i^{(n)}] = 0, \tag{13}$$

*where $\varepsilon_i^{(n)}$ is symmetric noise around zero. Define*

$$p_i := \Pr\left[\text{sign}(g_i^{(n)}) = \text{sign}(g_i)\right], \tag{14}$$

*where $p_i$ represents the probability that the sign of a single per-sample gradient $g_i^{(n)}$ matches the true gradient $g_i$.*

*Then, for all $i$ with $g_i \neq 0$, $p_i > 1/2$.*

*In particular, the majority-vote estimator*

$$\hat{s}_i := \text{sign}\left(-\sum_{n=1}^{N} \text{sign}(g_i^{(n)})\right) \tag{15}$$

*recovers the correct sign with probability at least*

$$\Pr[\hat{s}_i = \text{sign}(-g_i)] \geq 1 - \exp\left(-2N(p_i - 1/2)^2\right). \tag{16}$$

*Proof.* We divide the proof into two parts.

**Step 1: Bias of single-sample signs.**

Define the indicator random variable $X_n := \mathbb{1}\{\text{sign}(g_i^{(n)}) = \text{sign}(g_i)\} \in \{0, 1\}$. The success probability of a single sample is $p_i = \Pr[X_n = 1] = \Pr[\text{sign}(g_i^{(n)}) = \text{sign}(g_i)]$.

Without loss of generality, assume $g_i > 0$. The event of a successful sign match is $g_i^{(n)} > 0$, which can be rewritten as $g_i + \varepsilon_i^{(n)} > 0$, or $\varepsilon_i^{(n)} > -g_i$. Since the noise $\varepsilon_i^{(n)}$ is symmetric around zero, we know that $\Pr[\varepsilon_i^{(n)} > 0] = \Pr[\varepsilon_i^{(n)} < 0] = 1/2$. Because $g_i > 0$, the interval $(-g_i, 0)$ is non-empty. The probability of the noise falling into this interval, $\Pr[-g_i < \varepsilon_i^{(n)} < 0]$, is positive; therefore, the total probability of success is:

$$\begin{aligned} p_i &= \Pr[\varepsilon_i^{(n)} > -g_i] \\ &= \Pr[\varepsilon_i^{(n)} > 0] + \Pr[-g_i < \varepsilon_i^{(n)} < 0] \\ &= 1/2 + \Pr[-g_i < \varepsilon_i^{(n)} < 0]. \end{aligned}$$

Since $\Pr[-g_i < \varepsilon_i^{(n)} < 0] > 0$, it follows that $p_i > 1/2$.

**Step 2: Concentration of majority vote.**

The majority-vote estimator $\hat{s}_i$ succeeds if

$$\sum_{n=1}^{N} X_n > N/2. \tag{17}$$

Now, we bound the probability of failure, which is the event that the sum of correct sign estimates is less than or equal to $N/2$. We can express this event as a deviation from the expected value of the sum. The expected value of the sum is $\mathbb{E}\left[\sum_{n=1}^{N} X_n\right] = \sum_{n=1}^{N} \mathbb{E}[X_n] = Np_i$. Thus, the deviation is:

$$\sum_{n=1}^{N} X_n - \mathbb{E}\left[\sum_{n=1}^{N} X_n\right] = \sum_{n=1}^{N} X_n - Np_i \tag{18}$$

If we rewrite the event of failure, $\sum_{n=1}^{N} X_n \leq N/2$, in terms of this deviation we obtain:

$$\sum_{n=1}^{N} X_n - Np_i \leq N/2 - Np_i = -N(p_i - 1/2) \tag{19}$$

According to Hoeffding's inequality (Hoeffding, 1963) for a sum of i.i.d. random variables $X_n \in [0,1]$, we have:

$$\Pr\left(\sum_{n=1}^{N} X_n - Np_i \leq -N(p_i - 1/2)\right) \leq \exp\left(-\frac{2\left(N(p_i - 1/2)\right)^2}{\sum_{n=1}^{N}(1-0)^2}\right) \tag{20}$$

The denominator simplifies to $\sum_{n=1}^{N} 1^2 = N$. Substituting this back into the inequality gives:

$$\Pr\left[\sum_{n=1}^{N} X_n \leq N/2\right] \leq \exp\left(-\frac{2N^2(p_i - 1/2)^2}{N}\right) = \exp\left(-2N(p_i - 1/2)^2\right) \tag{21}$$

The probability of correct recovery is the complement of this failure probability:

$$\Pr[\hat{s}_i = \operatorname{sign}(-g_i)] = \Pr\left[\sum_{n=1}^{N} X_n > N/2\right] = 1 - \Pr\left[\sum_{n=1}^{N} X_n \leq N/2\right] \tag{22}$$

Therefore, we obtain the final bound:

$$\Pr[\hat{s}_i = \operatorname{sign}(-g_i)] \geq 1 - \exp\left(-2N(p_i - 1/2)^2\right). \tag{23}$$

$\square$

This result formalizes the intuition that, under mild assumptions on per-sample gradient noise, the majority-vote sign over a small batch of samples provides a reliable approximation to the true descent direction. The probability of correct recovery grows exponentially with both the number of samples $N$ and the signal-to-noise ratio $p_i - 1/2$. In practice, this suggests that even a few labeled samples can suffice to construct a gradient-sign mask that preserves most descent-aligned components of the source task vector.

# B ADDITIONAL RESULTS

## B.1 VISION RESULTS

Table 5: Cross-dataset performance of ViT-B/16 ($A$: `datacomp xl s13b b90k`, $B$: `laion2b s34b b88k`) models averaged across random seeds.

| Model | $|\mathcal{D}_s^c|$ | EUROSAT | RESISC45 | GTSRB | SVHN | DTD | AVG |
|---|---|---|---|---|---|---|---|
| $\theta_B$ zero-shot | | 49.41 | 67.98 | 48.29 | 50.58 | 55.96 | 54.45±7.30 |
| $\theta_B$ fine-tune | | 98.70 | 95.66 | 98.65 | 97.45 | 83.19 | 94.73±5.94 |
| $\theta_B + \tau_A$ | | 49.58 | 67.87 | 49.31 | 50.84 | 56.27 | 54.78±7.05 |
| $\theta_B + \delta^\star$ (oracle) | | 95.06 | 87.06 | 82.92 | 92.04 | 71.44 | 85.71±8.30 |
| $\theta_B^{opt}$ | 1 | 56.61±6.06 | 69.25±0.96 | 56.08±2.87 | 61.32±4.09 | 56.21±0.88 | 59.89±6.05 |
| $\theta_B + \delta^A$ | 1 | 61.94±0.43 | 70.05±0.56 | 60.88±2.85 | 71.07±1.82 | 58.32±0.30 | 64.45±5.47 |
| $\theta_B^{opt}$ | 2 | 59.49±1.43 | 71.20±1.13 | 61.70±0.76 | 62.01±4.40 | 57.00±0.54 | 62.29±5.31 |
| $\theta_B + \delta^A$ | 2 | 65.07±1.10 | 71.42±0.90 | 64.33±1.05 | 70.19±4.55 | 58.51±0.10 | 65.96±5.06 |
| $\theta_B^{opt}$ | 5 | 61.99±7.29 | 73.01±0.48 | 63.08±1.41 | 67.03±3.93 | 59.65±0.80 | 64.95±5.81 |
| $\theta_B + \delta^A$ | 5 | 66.05±1.21 | 71.57±0.88 | 66.61±0.42 | 73.59±0.82 | 60.02±0.20 | 67.57±4.96 |
| $\theta_B^{opt}$ | 10 | 59.98±3.77 | 72.27±1.65 | 64.54±1.12 | 67.85±1.02 | 60.96±0.33 | 65.12±4.97 |
| $\theta_B + \delta^A$ | 10 | 66.59±1.83 | 72.05±0.59 | 66.02±1.59 | 74.82±1.22 | 60.18±0.40 | 67.93±5.38 |
| $\theta_B^{opt}$ | 20 | 60.59±3.94 | 74.22±0.66 | 65.51±0.80 | 67.19±0.65 | 62.59±0.08 | 66.02±5.10 |
| $\theta_B + \delta^A$ | 20 | 67.05±0.41 | 72.29±0.14 | 66.42±0.47 | 74.11±0.59 | 60.92±0.08 | 68.15±4.84 |
| $\theta_B^{opt}$ | 50 | 58.80±2.45 | 75.88±0.83 | 64.91±0.53 | 67.58±2.91 | 64.13±0.11 | 66.26±5.97 |
| $\theta_B + \delta^A$ | 50 | 66.94±0.46 | 72.26±0.22 | 66.13±0.14 | 74.07±1.52 | 61.35±0.03 | 68.15±4.75 |

Table 6: Cross-dataset performance of ViT-L/14 ($A$: `datacomp xl s13b b90k`, $B$: `laion2b s32b b82k`) models averaged across random seeds.

| Model | $|\mathcal{D}_s^c|$ | EUROSAT | RESISC45 | GTSRB | SVHN | DTD | AVG |
|---|---|---|---|---|---|---|---|
| $\theta_B$ zero-shot | | 62.80 | 73.12 | 56.12 | 37.28 | 63.35 | 58.53±12.35 |
| $\theta_B$ fine-tune | | 98.95 | 97.06 | 99.16 | 97.80 | 83.56 | 95.31±6.13 |
| $\theta_B + \tau_A$ | | 62.77 | 73.49 | 56.03 | 39.09 | 63.56 | 58.99±11.80 |
| $\theta_B + \delta^\star$ (oracle) | | 96.75 | 90.30 | 88.65 | 92.60 | 72.66 | 88.19±8.52 |
| $\theta_B^{opt}$ | 1 | 64.65±5.90 | 74.54±0.57 | 63.97±4.50 | 62.51±4.58 | 63.76±0.27 | 65.89±5.61 |
| $\theta_B + \delta^A$ | 1 | 69.67±1.44 | 76.45±1.33 | 66.82±0.84 | 70.15±5.18 | 65.50±0.74 | 69.72±4.46 |
| $\theta_B^{opt}$ | 2 | 70.76±1.77 | 76.62±0.26 | 69.91±1.89 | 45.23±1.87 | 64.97±0.21 | 65.50±11.23 |
| $\theta_B + \delta^A$ | 2 | 74.10±2.00 | 76.97±0.68 | 71.55±2.73 | 54.31±2.54 | 66.10±0.59 | 68.61±8.44 |
| $\theta_B^{opt}$ | 5 | 69.75±1.64 | 75.41±2.94 | 73.25±0.62 | 67.11±2.39 | 66.72±0.13 | 70.45±3.86 |
| $\theta_B + \delta^A$ | 5 | 75.59±2.24 | 76.82±0.48 | 73.14±1.23 | 74.41±2.22 | 66.95±0.69 | 73.31±3.88 |
| $\theta_B^{opt}$ | 10 | 70.36±3.82 | 78.07±1.76 | 74.11±0.31 | 60.43±3.68 | 69.36±0.27 | 70.46±6.45 |
| $\theta_B + \delta^A$ | 10 | 73.74±1.58 | 77.59±0.57 | 74.94±0.73 | 75.88±2.80 | 67.41±0.48 | 73.77±3.86 |
| $\theta_B^{opt}$ | 20 | 78.74±2.96 | 80.77±1.17 | 74.65±0.28 | 65.99±2.12 | 71.40±0.35 | 74.31±5.65 |
| $\theta_B + \delta^A$ | 20 | 74.87±0.71 | 78.16±0.33 | 74.90±0.55 | 75.79±0.90 | 67.55±0.24 | 74.15±3.83 |
| $\theta_B^{opt}$ | 50 | 77.07±1.66 | 82.16±0.31 | 75.38±0.45 | 67.83±1.64 | 73.81±0.08 | 75.25±4.90 |
| $\theta_B + \delta^A$ | 50 | 74.75±0.89 | 78.27±0.18 | 74.61±0.91 | 76.79±1.26 | 67.77±0.01 | 74.27±3.86 |

# C  ABLATIONS

## C.1  SIGN AGREEMENT ANALYSIS

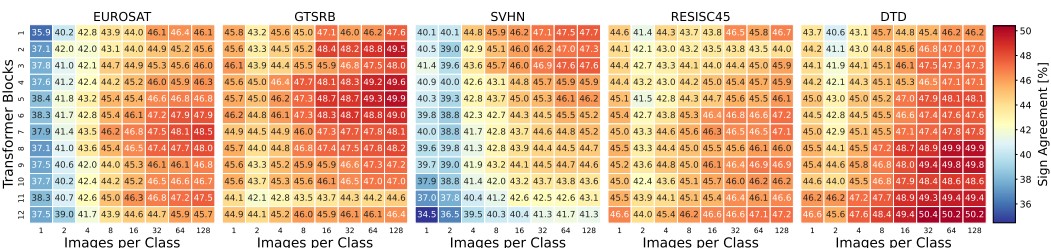

Figure 3: Sign agreement per block for ViT-B/16. The heatmaps show the percentage of sign agreements between the $m^\star$ and $m$ constructed by computing the signs of the anti-gradients at different $|\mathcal{D}_s^c|$ budgets.

To understand the nature of the misalignment between source and target models, we show in Fig. 3 a heatmap of anti-gradient sign agreement. Specifically, we computed the element-wise agreement between the signs of the target task vector ($\tau_B$) and the estimated anti-gradients at different data budgets.

This heatmap reveals that agreement is not uniform across different layers, and there is no simple, global sign correlation. This observation provides a direct explanation for the failure of naive task vector transfer. Without a mechanism to correct for this misalignment, simply adding the task vector introduces harmful directions that degrade performance. Our method, in contrast, actively addresses this by using the target model's gradients to construct a mask, ensuring that only the useful components of the task vector are transferred and aligned with the target loss landscape. Tables 5 and 6 demonstrate that $\delta^A$ consistently matches or outperforms $\theta_B^{opt}$ across both ViT-B/16 and ViT-L/14 models, even at larger $|\mathcal{D}_s^c|$ budgets. Moreover, $\delta^A$ exhibits substantially lower standard deviation across seeds, confirming its robustness to the random choice of supervision data and its efficiency compared to direct fine-tuning under identical data constraints.

## C.2  SUBSET DATA SELECTION

In the main experiments, we randomly selected the subset $\mathcal{D}_s$ from $\mathcal{D}$. As an ablation study, we now evaluate different heuristics—random, herding, $k$-medoids, and coreset—for constructing $\mathcal{D}_s$ to estimate anti-gradient signs, aiming to understand how subset selection impacts the accuracy and efficiency of anti-gradient estimation. Each strategy is evaluated at $b \in \{1, 2, 5, 10, 20\}$ examples per class. For herding (Rebuffi et al., 2016; Harvey, 2014), $k$-medoids (Kaufman & Rousseeuw, 1987), and coreset (Sener & Koltun, 2018), images are embedded using the frozen CLIP image encoder of the source model $\theta_A$. Let $f$ denote this frozen image encoder, normalized features are computed as $\boldsymbol{z}(x) = f(x)/\|f(x)\|$, and let $\mathcal{D}^c := \{(x, y) \in \mathcal{D} \mid y = c\}$.

**Random.** Sample uniformly from $\mathcal{D}$ without replacement.

**Herding.** Greedily select representatives $\mathcal{D}_s^c \subseteq \mathcal{D}^c$ of size $b$ to match the class mean feature by minimizing the discrepancy of the running average:

$$\mathcal{D}_s^c = \arg\min_{|\mathcal{D}_s^c|=b} \left\| \boldsymbol{\mu}_c - \frac{1}{|\mathcal{D}_s^c|} \sum_{x \in \mathcal{D}_s^c} \boldsymbol{z}(x) \right\|_2, \qquad \boldsymbol{\mu}_c := \frac{1}{|\mathcal{D}^c|} \sum_{x \in \mathcal{D}^c} \boldsymbol{z}(x). \tag{24}$$

$k$**-Medoids.** Select $\mathcal{D}_s^c \subseteq \mathcal{D}^c$ of size $b$ that minimizes the in-class assignment cost under distance $d$:

$$\mathcal{D}_s^c = \arg\min_{|\mathcal{D}_s^c|=b} \sum_{x \in \mathcal{D}^c} \min_{s \in \mathcal{D}_s^c} d\big(\boldsymbol{z}(x), \boldsymbol{z}(s)\big). \tag{25}$$

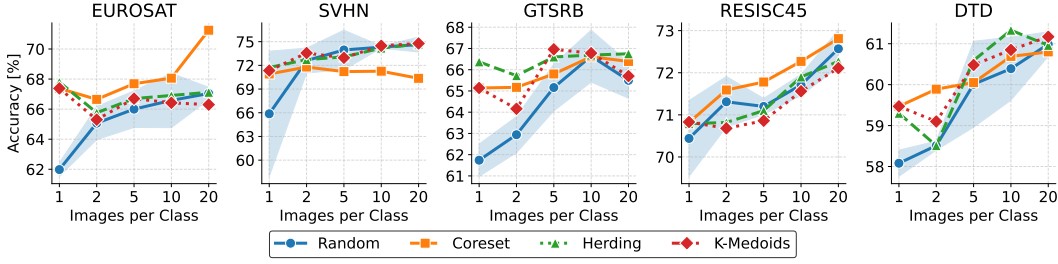

Figure 4: Accuracy across different numbers of images per class for various $\mathcal{D}_s$ construction heuristics.

**Coreset (medoid-proximity greedy).** Adopt a medoid-proximity greedy selection within $\mathcal{D}^c$:

$$
\begin{aligned}
\text{(i) Seed:} \quad & s_1 = \underset{j \in \mathcal{D}^c}{\arg\min} \sum_{k \in \mathcal{D}^c} d\left(\boldsymbol{z}(j), \boldsymbol{z}(k)\right), \\
\text{(ii) Greedy:} \quad & s_t = \underset{j \in \mathcal{D}^c \setminus \mathcal{D}^c_{s,t-1}}{\arg\min} \; \underset{s \in \mathcal{D}^c_{s,t-1}}{\min} d\left(\boldsymbol{z}(j), \boldsymbol{z}(s)\right), \quad t = 2, \dots, b,
\end{aligned}
\tag{26}
$$

where $\mathcal{D}^c_{s,t-1} = \{s_1, \dots, s_{t-1}\}$ denotes the set of already selected samples. This strategy emphasizes prototypical samples to reduce variance in small budgets and is closely related to coreset selection for active learning (Sener & Koltun, 2018).

For each dataset and budget $b$, $\mathcal{D}_s = \bigcup_c \mathcal{D}^c_s$, anti-gradient signs are estimated as discussed in Sec. 4.3 using majority-vote aggregation, and masked transport is applied to $\theta_B$. We report in Fig. 4 the accuracy for each strategy as a function of images per class, with standard deviation across random seeds for the random baseline. Across datasets and budgets, structured selectors (coreset, herding, $k$-medoids) often provide small but consistent gains over random selection in the few-shot regime. Yet, random selection remains a strong baseline, with performance approaching that of structured methods as $b$ increases, while incurring no memory or computation overhead from embedding or distance pre-computation. Importantly, this shows that our approach remains effective even when the subset $\mathcal{D}_s$ is chosen at random, validating its applicability in strict few-shot settings where sophisticated selection strategies are infeasible. Trends are stable across seeds, with variance shrinking as $b$ grows. Finally, structured methods require access to the full target dataset $\mathcal{D}$, which is unrealistic in privacy-constrained or large-scale settings.

## C.3 RANDOM TASK VECTOR TRANSPORT

Table 7: Ablation study comparing ViT-B/16 models with GradFix ($\delta^A$) against a randomized task vector baseline ($\delta^{\text{random}}$). The random vector preserves the mean and standard deviation of the source task vector but lacks its structural information.

| Model | $|\mathcal{D}_s^c|$ | EUROSAT | SVHN | GTSRB | RESISC45 | DTD | AVG |
|---|---|---|---|---|---|---|---|
| $\theta_B$ zero-shot | - | 49.41 | 50.58 | 48.29 | 67.98 | 55.96 | 54.44 |
| $\theta_B$ fine-tune | - | 98.70 | 97.45 | 98.65 | 95.66 | 83.19 | 94.73 |
| $\theta_B + \tau_A$ | - | 49.58 | 50.84 | 49.31 | 67.87 | 56.27 | 54.77 |
| $\theta_B + \delta^\star$ | - | 95.06 | 92.04 | 82.92 | 87.06 | 71.44 | 85.70 |
| $TransFusion$ | - | 50.12 | 53.26 | 50.24 | 67.99 | 56.70 | 55.66 |
| $\theta_B^{opt}$ | 1 | 56.61 | 61.32 | 56.08 | 69.25 | 56.21 | 59.89 |
| $\theta_B + \delta^{\text{random}}$ | 1 | 49.86 | 51.43 | 48.56 | 68.06 | 56.06 | 54.79 |
| $\theta_B + \delta^A$ | 1 | 61.94 | 71.07 | 60.88 | 70.05 | 58.32 | 64.45 |
| $\theta_B^{opt}$ | 2 | 59.49 | 62.01 | 61.70 | 71.20 | 57.00 | 62.28 |
| $\theta_B + \delta^{\text{random}}$ | 2 | 50.07 | 51.45 | 48.64 | 68.14 | 56.05 | 54.87 |
| $\theta_B + \delta^A$ | 2 | 65.07 | 70.19 | 64.33 | 71.42 | 58.51 | 65.90 |
| $\theta_B^{opt}$ | 5 | 61.99 | 67.03 | 63.08 | 73.01 | 59.65 | 64.95 |
| $\theta_B + \delta^{\text{random}}$ | 5 | 50.24 | 51.79 | 48.65 | 68.11 | 57.74 | 55.31 |
| $\theta_B + \delta^A$ | 5 | 66.05 | 73.59 | 66.61 | 71.57 | 60.02 | 67.57 |

To assess whether the effectiveness of GradFix arises from the transfer of meaningful structural knowledge from the source task vector $\tau_A$, rather than solely from the gradient masking mechanism, we performed an ablation using a random task vector. Specifically, we replaced the original source vector $\tau_A$ with a randomly generated vector $\tau_{\text{random}}$, drawn from a normal distribution matched to the element-wise statistics of $\tau_A$ to ensure a fair comparison:

$$\tau_{\text{random}} \sim \mathcal{N}(\mu_{\tau_A}, \sigma_{\tau_A}^2) \tag{27}$$

where $\mu_{\tau_A}$ and $\sigma_{\tau_A}$ are the mean and standard deviation of the parameters in $\tau_A$. We then applied our proposed gradient-sign masking method to this random vector using the same target gradient mask $\boldsymbol{m}$. The resulting update is defined as:

$$\theta_B^{\text{rand}} = \theta_B + \alpha(\boldsymbol{m} \odot \tau_{\text{random}}) \tag{28}$$

We evaluated this baseline using supervision budgets of $|\mathcal{D}_s^c| \in \{1, 2, 5\}$ samples per class.

The results are reported in Tab. 7. Notably, the performance of the random vector is often comparable to, or only marginally better than, the zero-shot baseline ($\theta_B$). These findings confirm that the gradient-sign mask $\boldsymbol{m}$ alone is insufficient to induce strong performance; it must act on a vector that contains valid task-specific information, as provided by $\tau_A$.

## C.4 GENERALIZATION AND ROBUSTNESS

To ensure that merging the transported task vector does not compromise the robust capabilities of the new backbone, we evaluated performance on a held-out support set, **ImageNet-R** (Hendrycks et al., 2020). This allows us to assess whether the target model's superior zero-shot capabilities are preserved after transfer.

As shown in Tab. 8, transferring the task vector via GradFix ($\theta_B + \delta^A$) substantially improves accuracy on the downstream tasks compared to the zero-shot baseline, while fully preserving the superior zero-shot generalization of the target model. This confirms that our method effectively adapts the model to the target task without sacrificing the general robustness of the updated pre-training.

Table 8: **Generalization analysis.** We report accuracy on each downstream task and on the held-out ImageNet-R support set. GradFix improves task performance while preserving the target model's zero-shot capabilities.

| Method | EUROSAT | | SVHN | | GTSRB | | RESISC45 | | DTD | |
|---|---|---|---|---|---|---|---|---|---|---|
| | Task | Supp. | Task | Supp. | Task | Supp. | Task | Supp. | Task | Supp. |
| $\theta_A$ zero-shot | 49.08 | 68.80 | 47.00 | 68.80 | 43.37 | 68.80 | 58.94 | 68.80 | 47.50 | 68.80 |
| $\theta_B$ zero-shot | 49.41 | 79.78 | 50.58 | 79.78 | 48.29 | 79.78 | 67.98 | 79.78 | 55.96 | 79.78 |
| $\theta_A$ ft | 98.58 | 65.93 | 93.59 | 67.77 | 98.23 | 65.98 | 92.43 | 65.35 | 79.04 | 65.02 |
| $\theta_B + \delta^A$ | 67.80 | 79.77 | 69.74 | 79.73 | 65.23 | 80.17 | 72.14 | 79.88 | 59.63 | 79.27 |

## D DATASETS AND SUPERVISION PROPORTIONS

In this section, we provide detailed information about the datasets used in our experiments and compute, for each one, the supervision proportions corresponding to our subset budgets $|\mathcal{D}_s^c|$. Recall that $|\mathcal{D}_s^c|$ denotes the number of labeled examples *per class* used to estimate anti-gradient signs. The resulting percentages indicate what fraction of the full training set those few-shot budgets represent.

### D.1 VISUAL DATASETS

- **EuroSAT**: A dataset based on Sentinel-2 satellite images covering 13 spectral bands, consisting of 27 000 labeled and geo-referenced samples across 10 classes (Helber et al., 2019).

- **SVHN**: A real-world image dataset from Google Street View house numbers, containing 73 257 labeled digits across 10 classes (Netzer et al., 2011).

- **GTSRB**: The German Traffic Sign Recognition Benchmark, comprising 39 209 training images and 12 630 test images across 43 classes (Stallkamp et al., 2011).

- **RESISC45**: A scene classification dataset with 31 500 RGB images $256 \times 256$ from Google Earth, covering 45 scene classes with 700 images per class (Cheng et al., 2017).

- **DTD**: The Describable Textures Dataset, consisting of 5640 images organized into 47 categories inspired by human perception (Cimpoi et al., 2014).

Table 9: Supervision proportions for visual datasets. $|\mathcal{D}_s^c|$ denotes examples per class. Each cell shows the total dataset percentage.

| Dataset | # Samples | Classes | $|\mathcal{D}_s^c|$ | | | | | |
|---|---|---|---|---|---|---|---|---|
| | | | 1 | 2 | 5 | 10 | 20 | 50 |
| EUROSAT | 27,000 | 10 | 0.04% | 0.07% | 0.19% | 0.37% | 0.74% | 1.85% |
| SVHN | 73,257 | 10 | 0.01% | 0.03% | 0.07% | 0.14% | 0.27% | 0.68% |
| GTSRB | 39,209 | 43 | 0.11% | 0.22% | 0.55% | 1.10% | 2.19% | 5.48% |
| RESISC45 | 31,500 | 45 | 0.14% | 0.29% | 0.71% | 1.43% | 2.86% | 7.14% |
| DTD | 5,640 | 47 | 0.83% | 1.66% | 4.15% | 8.30% | 16.60% | 41.49% |

### D.2 TEXTUAL DATASETS

- **SNLI**: Stanford Natural Language Inference dataset, containing 570 000 sentence pairs labeled for entailment, contradiction, or neutral (Group et al., 2022).

- **MNLI**: Multi-Genre Natural Language Inference dataset, comprising 433 000 sentence pairs annotated with textual entailment information across various genres (Williams et al., 2018).

- **RTE**: Recognizing Textual Entailment dataset, with 2490 examples for training, 277 for validation, and 3000 for testing, divided into two classes (Wang et al., 2018).

- **QNLI**: Question Natural Language Inference dataset, containing 104 743 training examples divided into two classes (Wang et al., 2018).

- **SCITAIL**: A science entailment dataset built from science question answering, with 23 596 training examples divided into two classes (Khot et al., 2018).

Table 10: Supervision proportions for textual datasets. $|\mathcal{D}_s^c|$ denotes examples per class. Each cell shows the total dataset percentage.

| Dataset | # Samples | Classes | $|\mathcal{D}_s^c| = 50$ |
|---|---|---|---|
| SNLI | 570,000 | 3 | 0.03% |
| MNLI | 433,000 | 3 | 0.03% |
| RTE | 2,490 | 2 | 4.02% |
| QNLI | 104,743 | 2 | 0.10% |
| SCITAIL | 23,596 | 2 | 0.42% |

## E  HYPERPARAMETER SELECTION

We evaluated the optimal task vector application coefficient $\alpha$ using the validation set of each dataset, following standard practice (Ilharco et al., 2023; Gargiulo et al., 2025; Marczak et al., 2025). For $\delta^\star$, the optimal $\alpha$ is equal to 1 across all datasets. In Tab. 11, we summarize the coefficients corresponding to the optimal performance of $\delta^A$ for each dataset. $\theta_B^{\text{opt}}$ is obtained for each dataset by training with the AdamW optimizer (learning rate 1e−5) on the corresponding subset $\mathcal{D}_s$ used to compute the gradient mask $m$, with a single gradient-descent step.

| Architecture | EUROSAT | SVHN | GTSRB | RESISC45 | DTD |
|---|---|---|---|---|---|
| ViT-B/16 | 0.3 | 0.3 | 0.6 | 0.2 | 0.7 |
| ViT-L/14 | 0.4 | 0.5 | 0.5 | 0.2 | 0.5 |
| **Architecture** | **SNLI** | **MNLI** | **RTE** | **QNLI** | **SCITAIL** |
| T5v1.1 | 0.3 | 0.1 | 0.2 | 0.1 | 0.7 |

Table 11: Optimal hyperparameters for $\delta^A$ across CLIP and T5 architectures.

## F  COMPUTATIONAL COST ANALYSIS

Following standard scaling-law approximations (Kaplan et al., 2020), we adopt the commonly used per-parameter FLOP estimates to compare the computational efficiency of our approach against baselines. Let $P$ denote the number of model parameters. We assume the following operation costs:

- **1 forward pass:** $\approx 2P$ FLOPs
- **1 backward pass:** $\approx 4P$ FLOPs
- **Optimizer update (Adam/AdamW):** $\approx 10P$ FLOPs per step (accounting for momentum updates, squaring, bias correction, and parameter write-back).

Given this, we derive the costs for GradFix, the few-shot baseline $\theta_B^{opt}$, and full fine-tuning.

**GradFix cost.** GradFix requires a single gradient computation on the target model followed by the masking operation.

- Forward + Backward pass: $6P$ FLOPs
- Mask construction and masked task-vector application (element-wise multiplication and addition): $\approx 2P$ FLOPs

Total GradFix Cost: $\approx 8P$ FLOPs.

$\theta_{\mathbf{B}}^{\mathbf{opt}}$ **cost (one-step fine-tuning).** The $\theta_B^{opt}$ baseline represents a one-step fine-tuning update on the target model.

- Forward + Backward pass: $6P$ FLOPs
- Adam optimizer update: $10P$ FLOPs

Total $\theta_B^{opt}$ Cost: $\approx 16P$ FLOPs.

**Full fine-tuning cost.** Standard fine-tuning (used to produce the original source task vectors) typically involves 2000 training steps.

- Per step (Forward + Backward + Optimizer): $6P + 10P = 16P$ FLOPs
- For 2000 steps: $16P \times 2000$

Comparing the methods reveals significant efficiency gaps:

- $\theta_B^{opt}$ vs. GradFix: $\approx 2\times$ more FLOPs
- Full fine-tuning vs. GradFix: $\approx 4,000\times$ more FLOPs
- Full fine-tuning vs. $\theta_B^{opt}$: $\approx 2,000\times$ more FLOPs

Euro

