# OpenReview forum: "Gradient-Sign Masking for Task Vector Transport Across Pre-Trained Models"
_ICLR.cc/2026/Conference — ICLR 2026 Poster_

### Official Review · Reviewer_VKR3 · 2025-11-01

**Soundness:** 3
**Presentation:** 3
**Contribution:** 3
**Rating:** 6
**Confidence:** 3

**Summary:**

This paper presents GradFix, which aims to transfer task vectors between different pre-trained models by leveraging gradient sign structures in settings with limited downstream data. Specifically, GradFix computes gradients on the target pre-trained model and uses the resulting gradient signs to mask out harmful update directions that may increase the downstream loss. The paper also provides a theoretical guarantee showing that this procedure yields a first-order reduction in loss. To address the limited data regime, a majority-vote strategy is adopted to robustly estimate gradient signs. Experimental results demonstrate that GradFix outperforms baselines such as naive transfer and related approaches, and reduces the performance gap to full-dataset fine-tuning.

**Strengths:**

- Overall, the paper is well-written and easy to follow. It investigates an important and practical problem regarding improving the transfer efficiency of task vectors amid the rapid advancement of foundation models. Moreover, the simplicity of the proposed method also makes it easy to understand and implement.

- The experimental results are promising, showing that GradFix consistently outperforms the baselines and related methods, and reduces the performance gap to full-dataset fine-tuning.

- The paper provides a theoretical guarantee that GradFix ensures a first-order decrease in the loss. This also helps explain why and how the proposed method is effective.

- Experiments examine the behavior of GradFix under different types of downstream data subsets and hyperparameter settings, which helps improve the understanding of the method’s applicability.

**Weaknesses:**

- At Lines 36-38, the paper states that one motivation is to reduce redundant re-training when new models are released by companies or researchers. However, newer and more advanced models often involve architectural changes (e.g., increased model size to accommodate knowledge learned from larger datasets), whereas GradFix appears to require the pre-trained models to share the same architecture. This constraint may limit the applicability of the method in such scenarios.

- At Lines 469-471, the paper states that GradFix does not require fine-grained tuning of $\alpha$. However, Figure 3 indicates that different downstream datasets achieve optimal performance at different $\alpha$ values. For example, $\alpha = 0.3$ performs best for EUROSAT, SVHN, and RESISC45, whereas $\alpha = 0.7$ is optimal for GTSRB. Moreover, the accuracy differences across $\alpha$ values on these datasets can be as large as 10-20%. Despite this, the paper does not provide a strategy for selecting $\alpha$ when adapting to new datasets or pre-trained models in practical use cases.

- Table 1 and Table 2 do not report accuracy variance across different sample selections in the few-shot setting. However, Figure 2 suggests that random data selection can still affect the performance of GradFix to a certain degree. It is therefore unclear how the performance variance of GradFix compares to that of $\theta_B^{\text{opt}}$​. For example, are there few-shot sample selections where $\theta_B^{\text{opt}}$​ actually outperforms GradFix, and how frequently does this occur?

**Questions:**

Besides the weakness shown in the above section, please also see the following questions:

Q1: From Equations (1) and (4), the mask $m$ seems to take binary values (0 or 1). However, Figure 1 depicts $m$ with positive and negative signs. Why does the figure illustration of $m$ differ from its binary formulation?

Q2: In the limited-data scenario, the performance gap between GradFix and the oracle (i.e., $\theta_B + \delta^*$) remains substantial. How much does GradFix improve as more samples become available, for example, when the full dataset is used? Additionally, at what sample size does standard fine-tuning begin to outperform GradFix? Understanding this transition point would help clarify which approach is preferable under different computation, labeling, or storage budgets.

---

> ### Author Response · Authors · 2025-11-20
>
> ## Same-Architecture Assumption
> > At Lines 36-38, the paper states one motivation is reducing redundant re-training… However, newer models often involve architectural changes… GradFix appears to require pre-trained models to share the same architecture, which may limit its applicability.
>
>
> The reviewer is correct that GradFix, in its current form, assumes that the source and target models share the same architecture.
> This design choice was deliberate, allowing us to focus on one of the two core challenges in transferring task vectors:
>
> 1. **Architectural Alignment:** How can we map parameters from model $A$ to model $B$ when their architectures differ (e.g., different layer widths, increased model size, or structural modifications)?
> 2. **Knowledge Transfer:** Once aligned, how can we adapt the source task vector $\tau_A$ to the target model $\theta_B$, ensuring that it leverages the pre-trained knowledge effectively and remains a valid descent direction for the task?
>
> Our work addresses the second problem: the same-architecture assumption simplifies the experimental setting and corresponds to many real-world scenarios, such as upgrading from `T5-v1.0` → `T5-v1.1` or `CLIP-ViT-B-16` → `OpenCLIP-ViT-B-16`. In these cases, models share the same core architecture while benefiting from improvements in pre-training, making GradFix immediately applicable.
> We fully agree that extending GradFix to handle architectural differences, such as larger or structurally modified models, is an important future direction. Nonetheless, solving the same-architecture case remains a relevant contribution in its own right.
>
> ## Alpha selection strategy
> > At Lines 469-471, the paper states GradFix does not require fine-grained tuning of $\alpha$… However, Figure 3 shows different datasets perform best at different $\alpha$ values (e.g., 0.3 for EuroSAT, SVHN, RESISC45; 0.7 for GTSRB)… Accuracy differences across $\alpha$ can reach 10–20%… No strategy is provided for selecting $\alpha$ in new datasets or models.
>
> Determining the optimal magnitude for parameter updates is a known challenge, and existing methods such as Task Arithmetic (Ilharco et al., 2023), TIES-Merging (Yadav et al., 2023), KnOTS (Stoica et al., 2024), TSV (Gargiulo et al., 2025), Iso-C (Marczak et al., 2025), MagMax (Marczak et al., 2024), Core Space (Panariello et al., 2025), AdaMerging (Yang et al., 2024), and DARE (Yu et al., 2024) typically rely on hyperparameter sweeps on held-out validation sets to select this value.
>
> Our statement at Lines 469–471 refers to the reduced sensitivity of GradFix to the exact value of the scaling parameter, rather than claiming that no selection is required. As illustrated in **Figure 2**, Majority Voting consistently outperforms the Mean strategy and maintains higher accuracy over a broader range of $\alpha$. This stability arises because Majority Voting is less sensitive to variance and outliers in the small sample gradients compared to simple averaging, guaranteeing that the estimated sign concentrates rapidly around the true direction.
>
> This inherent stability means that while fine-tuning $\alpha$ yields the absolute theoretical optimum, precise tuning is not strictly necessary to achieve significant performance gains.
> To provide a practical strategy for few-shot or low-data scenarios where a validation set is unavailable, we follow the suggestion from another reviewer and adopt a fixed $\alpha = 0.5$ as a robust default.
>
> The following tables demonstrate that using this untuned $\alpha=0.5$ still ensures effective transfer, consistently outperforming the Naive Transfer baseline ($\theta_B + \tau_A$):
>
> **ViT-B/16 Results**
> | Dataset | Naive Transfer ($\theta_B + \tau_A$) | GradFix | Gain |
> | :--- | :--- | :--- | :--- |
> | **EuroSAT** | 49.14% | 60.63% | +11.49% |
> | **SVHN** | 50.62% | 65.37% | +14.75% |
> | **GTSRB** | 55.12% | 61.37% | +6.25% |
> | **RESISC45**| 65.57% | 68.33% | +2.76% |
> | **DTD** | 56.10% | 58.40% | +2.30% |
>
> **ViT-L/14 Results**
> | Dataset | Naive Transfer ($\theta_B + \tau_A$) | GradFix | Gain |
> | :--- | :--- | :--- | :--- |
> | **EuroSAT** | 62.33% | 69.50% | +7.17% |
> | **SVHN** | 38.82% | 73.26% | +34.44% |
> | **GTSRB** | 55.75% | 66.99% | +11.24% |
> | **RESISC45**| 72.43% | 77.80% | +5.37% |
> | **DTD** | 62.51% | 65.21% | +2.70% |
>
> These results confirm that the $\alpha=0.5$ heuristic is highly effective in practice, validating the robustness of GradFix's gradient-sign masking mechanism even when the ideal $\alpha$ is unknown.

---

> > ### Author Response · Authors · 2025-11-20
> >
> > ## Variance and Comparison with $\theta_B^{opt}$
> > > Table 1 and Table 2 do not report accuracy variance… Figure 2 suggests random data selection can affect GradFix’s performance… It is unclear how its variance compares to $\theta_B^{\text{opt}}$, e.g., are there few-shot samples where $\theta_B^{\text{opt}}$ outperforms GradFix, and how often?
> >
> >
> > While the main text focused on mean performance, we have provided a detailed breakdown of variance and stability across different sample selections in **Appendix B.1** (old B.3).
> >
> > Specifically, **Tables 6 and 7** demonstrate that GradFix consistently exhibits lower variance than the few-shot baseline ($\theta_B^{opt}$), indicating greater robustness to random data selection.
> >
> > ## Discrepancy between binary mask equations and Figure 1
> > > From Equations (1) and (4), the mask $m$ seems to take binary values (0 or 1). However, Figure 1 depicts $m$ with positive and negative signs. Why does the figure illustration of $m$ differ from its binary formulation?
> >
> > **Figure 1** is a schematic illustration intended to visualize the **sign agreement process** rather than the final binary mask values. In the figure, the positive ($+$) and negative ($-$) signs represent the **gradient signs** of the target model being compared against the source task vector.
> > * A match (e.g., $+$ and $+$) results in retaining the direction (mask value 1).
> > * A conflict (e.g., $+$ and $-$) results in suppressing the direction (mask value 0).
> >
> > We will clarify this distinction in the figure caption of the final revision to prevent confusion.
> >
> > ## Gap to Oracle and Crossover Point
> > > In the limited-data scenario, the performance gap between GradFix and the oracle ($\theta_B + \delta^*$) remains substantial… How much does GradFix improve with more samples, e.g., the full dataset?… At what sample size does standard fine-tuning outperform GradFix?… Understanding this transition clarifies which approach is preferable under different budgets.
> >
> >
> > To address the performance gap and the transition point where standard fine-tuning becomes preferable, we must distinguish between the nature of the vectors and the computational baselines used.
> >
> > **1. The Nature of the Gap**
> > The performance gap between GradFix and the oracle ($\tau_B$) arises from what these vectors represent:
> > * **A Fully-Finetuned Task Vector:** The "fully-finetuned counterpart" ($\tau_B$) is the result of a **multi-epoch optimization** process. The resulting task vector embeds a complex optimization path, influenced by learning rate schedules, momentum, and non-linear interactions across many epochs. Its sign structure represents a complete converged solution.
> > * **The GradFix Gradient Estimate:** Our method, GradFix, uses the gradient sign structure ($g_B$) computed from a **small budget of data**, often from a single forward-backward pass. This sign structure represents the initial noisy descent direction on the target model's loss landscape.
> > The gap essentially quantifies the difference between a local linear approximation (GradFix) and a converged global solution (Oracle).
> >
> > **2. Comparison Baseline and Transition Point**
> > To determine when fine-tuning becomes preferable, it is crucial to define the computational budget. Few-shot fine-tuning is conventionally interpreted as iterative training over multiple epochs. However, GradFix is designed as a "transport" mechanism rather than a training procedure. It requires only a single forward-backward pass to estimate gradient signs.
> > To isolate the specific contribution of the **transported task vector** (as opposed to information gained purely from the new data), we compare GradFix against a baseline ($\theta_B^{opt}$) constrained to the same computational budget. We have updated the paper to include a detailed computational cost comparison in **Appendix F**.
> > Under these constraints, **Tables 6 and 7** in the **Appendix B.1** reveal the transition point where learning solely from new data ($\theta_B^{opt}$) begins to outperform transporting the masked vector:
> >
> > **ViT-B/16:** GradFix remains superior even up to **50 shots per class** (68.15% vs. 66.26%).
> >
> > **ViT-L/14:** The crossover occurs between **10 and 20 shots**. At 10 shots, GradFix is superior (73.77% vs 70.46%), while at 20 shots, the baseline catches up (74.15% vs 74.31%)

---

### Official Review · Reviewer_sW3e · 2025-11-01

**Soundness:** 3
**Presentation:** 3
**Contribution:** 2
**Rating:** 4
**Confidence:** 4

**Summary:**

This paper addresses an interesting problem to transfer the task vector on a finetuned task across different pretrained checkpoints. The authors propose to mask the task vector based on whether its sign aligns with the gradient obtained with a small target set, which empirically improve the performance consistently.

Further ablations are needed to convincingly show that the observed improvement is due to cross-model task knowledge transfer rather than exploiting from a random vector based on incidental gradient sign alignment.

**Strengths:**

1. The proposed method is well-motivated, simple yet effective.
2. The method demonstrates consistent performance gain compared to zero-shot and other baselines.

**Weaknesses:**

1. While the proposed leads to consistent performance gain, the performance gap is still quite substantial compared to the fully-finetuned counterpart (e.g., 20%-40% accuracy gap). There seems to be a diminishing effect brought by increasing the size of the small target set to estimate the gradient sign (Table 5), which prevents the proposed method to eventually deliver on-par performance with fully-finetuned checkpoint even with large number of samples.

2. There is no ablation to verify whether improvements truly reflect transfer of task-specific knowledge from the original task vector or the model is just exploiting a random vector by incidental gradient alignment.

It is non-intuitive to readers that the original task vector can be transferred to the new checkpoint without any parameter permutation. One would expect the original task vector could be just random noise to the new pretrained checkpoint, which is partially verified by the results that naively adding them do not improve performance. Suppose it is actually noise, there is a possibility that the proposed method works because it is selecting the parameters which “luckily align with the right gradient direction” from the original task vector even though it is noise. I think a further ablation experiment is needed to verify whether this is the case. For example, I suggest the authors to test their proposed method on a randomly generated vector (e.g., a Gaussian vector with the same mean and std as actual task vector) and compare it with the performance obtained with the original task vector. The result comparison should verify to which extent the proposed method is exploiting from the actual task knowledge in the original task vector.

**Questions:**

Typos:
The alpha coefficient in Eq. 5 seems to be redundant, as it appears also in Eq. 6.

---

> ### Author Response · Authors · 2025-11-20
>
> ## Performance gap
> >While the proposed leads to consistent performance gain, the performance gap is still quite substantial compared to the fully-finetuned counterpart (e.g., 20%-40% accuracy gap). There seems to be a diminishing effect brought by increasing the size of the small target set to estimate the gradient sign (Table 5), which prevents the proposed method to eventually deliver on-par performance with fully-finetuned checkpoint even with large number of samples.
>
> The performance gap and the diminishing returns from increasing the sample size is expected and highlights a fundamental difference between the two quantities being compared:
>
> 1.  **A Fully-Finetuned Task Vector:** The "fully-finetuned counterpart"  is the result of a **multi-epoch optimization** process. The resulting task vector embeds a complex optimization path, influenced by learning rate schedules, momentum, and non-linear interactions across many epochs. Its sign structure represents a *converged solution*.
>
> 2.  **The Transported Task Vector:** Our method, GradFix, uses the gradient sign structure ($g_B$) computed from a **small budget of images** often from a single forward-backward pass. This sign structure represents the *initial descent direction* on the target model's loss landscape, and approximates the converged descent direction.
>
> Increasing the number of samples for our gradient estimation (as in Table 5) makes our estimate of this *initial gradient* more robust. It allows $g_B$ to better approximate the gradient of a single full-dataset epoch.
>
> However, the sign structure of a **single-epoch gradient** is fundamentally different from the sign structure of a **multi-epoch converged task vector**. Therefore, even with a large sample size, our gradient sign mask $m$ will not converge to the sign structure of a fully fine-tuned solution.
>
> This distinction is strongly supported by the **oracle experiment ($\theta_{B}+\delta^{*}$) in Table 1**. As noted by the Reviewer 1QUv, even when using a "perfect" oracle mask based on the target model's *own* fully fine-tuned task vector, the performance still lags behind a full fine-tune (e.g., EUROSAT: 95.06 vs 98.70; GTSRB: 82.92 vs 98.65). This demonstrates that the source task vector $\tau_A$ is fundamentally missing or contains conflicting information that even a perfect sign mask cannot fully overcome.
>
> Therefore, the observed performance gap is not a failure of the gradient estimation but rather a reflection of this fundamental information mismatch. GradFix is designed to successfully *transport* knowledge from $\tau_A$ by filtering it against the target's initial loss landscape, but it cannot (and is not designed to) recreate the new information that is only generated during a full multi-epoch fine-tuning process on the target model.
>
> ## Random baseline
> >There is no ablation to verify whether improvements truly reflect transfer of task-specific knowledge from the original task vector or the model is just exploiting a random vector by incidental gradient alignment.
>
> We performed an ablation using a randomly generated Gaussian vector matched to the original source task vector $\tau_A$ in **mean and standard deviation**. The results, reported in Supplementary **Section C.3** and **Table 8**, show that this random vector performs substantially worse than our method across all datasets and settings. This confirms that the improvements provided by GradFix arise from transferring meaningful task-specific knowledge encoded in the source task vector, rather than only from gradient-sign alignment. We thank the reviewer for this suggestion, as it strengthens the empirical validation of our approach.
>
> |Model|$\mathcal D_s^c$|EUROSAT|SVHN|GTSRB|RESISC45|DTD|AVG|
> |---|---|---|---|---|---|---|---|
> |$\theta_B$ zero-shot|-|49.41|50.58|48.29|67.98|55.96|54.44|
> |$\theta_B+\tau_A$|-|49.58|50.84|49.31|67.87|56.27|54.77|
> |
> |$\theta_B-\delta^A$|1|61.94|71.07|60.88|70.05|58.32|64.45|
> |$\theta_B-\delta^{rand}$|1|49.86|51.43|48.56|68.06|56.06|54.79|
> |
> |$\theta_B-\delta^A$|2|65.07|70.19|64.33|71.42|58.51|65.90|
> |$\theta_B-\delta^{rand}$|2|50.07|51.45|48.64|68.14|56.05|54.87|
> |
> |$\theta_B-\delta^A$|5|66.05|73.59|66.61|71.57|60.02|67.57|
> |$\theta_B-\delta^{rand}$|5|50.24|51.79|48.65|68.11|57.74|55.31|

---

> > ### Comment · Reviewer_sW3e · 2025-11-26
> >
> > Thank the authors for their clarification and the additional ablation experiment.
> >
> > My concern with the performance gap is not because it is not expected or it is not distinct from a fully fine-tuned vector. In fact, the fundamental difference between the proposed method and the fully fine-tuned vector is exactly my concern. As demonstrated by the results, the proposed method demonstrates a clear diminishing return with larger number of examples.
> >
> > In comparison, the few-shot fine-tuning baseline, which is currently outperformed by the proposed method in the low-data regime, is not limited by the same level of diminishing return, and will converge to the performance of a fully fine-tuned vector eventually with enough number of examples.
> >
> > Therefore, I raised the concern regarding the still-existing large performance gap with fully fine-tuned vector + this gap can not be filled even with sufficiently large number of samples. However, I also understand that the proposed method is designed for low-data regime and has computational effectiveness compared to fine-tuning.
> >
> > Based on the additional ablation which validates the effectiveness of knowledge transfer, I have updated my final score to 6.

---

### Official Review · Reviewer_G6x6 · 2025-11-05

**Soundness:** 3
**Presentation:** 3
**Contribution:** 2
**Rating:** 6
**Confidence:** 3

**Summary:**

The paper recognizes that updating checkpoints with new data or a new training pipeline leads to redundant repeated fine-tuning for the same downstream tasks. To address this, it proposes GradFix, which uses gradient-sign masking to transfer task-specific knowledge from the pre-trained model.

**Strengths:**

1.The paper proposes a fine-tuning-free knowledge transfer strategy that uses the sign of the gradient as a robust surrogate for the descent direction. The method is insightful and reasonably well-motivated.
2.The paper is clearly structured and well-organized, with high readability.

**Weaknesses:**

Please refer to the Questions section below.

**Questions:**

1.The proposed method appears applicable only to models with the same architecture trained on different downstream datasets, which limits its scope. If the two models share only similar architectures but differ in scale, does the method still apply?
2.The paper does not consider more complex multi-source transfer scenarios—i.e., simultaneously transferring multiple task vectors from different source models (such as models pre-trained on different datasets but adapted to the same task) into a single target model.
3.Can your method be transferred to diffusion-based generative models or VLM tasks? These pre-trained models are typically very large yet serve many downstream tasks and encode rich priors. If extending to VLM, what adaptations would be required?

---

> ### Author Response · Authors · 2025-11-20
>
> ## Cross-Scale Transfer
> >The proposed method appears applicable only to models with the same architecture trained on different downstream datasets, which limits its scope. If the two models share only similar architectures but differ in scale, does the method still apply?
>
> We thank the reviewer for raising this important point regarding the applicability of GradFix across models of different scales (e.g., transferring from `ViT-B` to `ViT-L`).
> In its current form, our method does not directly apply to models of different scales without an intermediate mapping step. This is because the source task vector $\tau_A$ and the target parameters $\theta_B$ reside in parameter spaces of different dimensions ($d_A \neq d_B$), making the element-wise operations defined in Eq. 4 and Eq. 5 impossible to perform immediately.
>
> However, the *principle* of GradFix remains highly relevant to the cross-scale setting. We view the challenge of cross-scale transport as consisting of two distinct sub-problems:
>
> 1.  **Dimensional Alignment:** Mapping parameters from source space $\mathbb{R}^{d_A}$ to target space $\mathbb{R}^{d_B}$, a challenge that is currently an open area of research.
> 2.  **Geometric Alignment:** Determining which components of the mapped vector align with the loss landscape of the target model to prevent negative transfer.
>
> Our paper focuses exclusively on solving the geometric alignment sub-problem. Even if one perfectly solves the dimensional alignment problem (resizing the weights), the resulting vector is not guaranteed to follow the descent direction of the target model due to the misaligned loss landscapes.
>
> To apply GradFix in a cross-scale setting, the workflow would be:
> 1.  **Map:** Compute a projected task vector $\tau_{A\to B} = \phi(\tau_A)$, where $\phi$ is a resizing function mapping dimensions from Model A to Model B.
> 2.  **Mask (GradFix):** Apply our method to the resized vector. Compute the gradient mask $m$ on $\theta_B$ and filter the projected vector: $\delta = m \odot \tau_{A}$.
>
> Thus, GradFix is fully compatible with cross-scale model transport once an alignment function $\phi$ is available. Our contribution focuses on geometric alignment which remains a requirement regardless of model scale.
>
> ## VLMs
> >Can your method be transferred to diffusion-based generative models or VLM tasks? These pre-trained models are typically very large yet serve many downstream tasks and encode rich priors. If extending to VLM, what adaptations would be required?
>
> The applicability of GradFix to Diffusion models and Vision-Language Models (VLMs) is feasible because our framework is mathematically agnostic to the specific architecture, relying only on the **differentiability of the loss objective** $\mathcal{L}$.
> In particular, our method relies on the premise that the gradient sign provides a robust surrogate for the local descent direction of the target loss landscape. Formally, the descent guarantee we provide in Eq. 6 and Eq. 7 (Section 4.2) relies on the first-order Taylor expansion:
>
> $\mathcal{L}(\theta_{B}-\alpha\delta^{A})\approx\mathcal{L}(\theta_{B})- g^{\top}\delta^{A}$
>
> This inequality holds for *any* differentiable loss function $\mathcal{L}$ and any parameterization $\theta$. Therefore, the masking mechanism defined in **Eq. 4**, which retains update components only when $sign(\tau_{A,i}) = sign(g_i)$ remains mathematically valid for ensuring the update does not increase the target loss by only retaining aligned components.
>
> Consequently, the method could potentially be extended to more complex architectures and objectives, such as:
> * **VLMs:** Since VLMs exploit components like Vision Encoders and Language Decoders, the possible applicability is supported by our experiments across these two modalities.
> * **Diffusion Models:** GradFix could be applied using the standard Denoising MSE loss. The gradient mask (Eq. 4) would align a source "concept vector" (e.g., a style-transfer task vector) with the target model trajectory, filtering updates that would increase reconstruction error.

---

> ### Author Response · Authors · 2025-11-20
>
> ## Multi-source transfer
> >The paper does not consider more complex multi-source transfer scenarios—i.e., simultaneously transferring multiple task vectors from different source models (such as models pre-trained on different datasets but adapted to the same task) into a single target model.
>
> To address it, we extended our experiments to a setting where task vectors from (K=5) different source models, fine-tuned on the same task but starting from different pre-trained backbones, are simultaneously transferred into a single target model $\theta_B$. This evaluation is reported in the newly added **Section 5.2** and **Table 4** of the revised paper.
> In this setting, direct merging (via Task Arithmetic or TIES) is geometrically invalid due to the misalignment of source parameter spaces, as evidenced by the catastrophic failure of the baselines. To resolve this, we employed a **Mask–then–Merge** strategy: we first transport each source vector individually to the target parameter space using GradFix, aligning them with the target's loss geometry, and subsequently merge the resulting updates. The results in the table below demonstrate that this strategy not only successfully recovers performance but, crucially, outperforms the single-source transport baseline reported in the main paper ($65.90\%$ avg). This indicates that GradFix enables the target model to benefit from an "ensemble" of descent directions derived from diverse source models, effectively smoothing out noise and producing a more robust update direction than a single transported vector.
>
> |Method|EURO|SVHN|GTSRB|RESISC|DTD|AVG|
> |:---|:--:|:--:|:--:|:--:|:--:|:--:|
> |$\theta_B$ zero-shot|49.41|50.58|48.29|67.98|55.96|54.44|
> |**TA Transport Merge**|||||||
> |TA baseline|36.94|35.09|30.77|50.54|44.73|39.61|
> |Mask→Merge|**65.96**|**72.97**|**65.80**|71.30|61.01|**67.41**|
> |**TIES Transport Merge**|||||||
> |TIES baseline|12.69|10.39|2.85|5.81|16.33|9.61|
> |Mask→Merge|65.17|72.58|65.36|**71.40**|**61.12**|67.13|
>
> We are grateful for this feedback, as these new findings significantly strengthen our paper by demonstrating the robustness of our approach in multi-source regimes.

---

### Official Review · Reviewer_PYpn · 2025-11-08

**Soundness:** 3
**Presentation:** 3
**Contribution:** 3
**Rating:** 4
**Confidence:** 4

**Summary:**

This paper proposes GradFix, a simple method to transfer task vectors between different pre-trained models. It masks the source task vector using the gradient sign of the target model, keeping only directions aligned with the target loss. This ensures a first-order descent update without re-training. Experiments on CLIP and T5 show that GradFix effectively transfers fine-tuning knowledge using only a few labeled samples, outperforming naive task-vector addition and few-shot tuning.

**Strengths:**

1. The idea is simple and easy to understand, yet surprisingly effective for transferring knowledge between models.
2. The method is lightweight and requires very little data, making it practical and efficient.
3. Results are clear and consistent across different tasks, showing strong generality and reliability

**Weaknesses:**

1. The motivation of the setting is not entirely convincing. If the source model has already been fine-tuned on the task, it is unclear why one would not simply use that model directly. Since obtaining the task vector requires access to the fine-tuned source, the need to reapply it to a target model feels less practical, especially when the transferred performance does not surpass the source.
2. The method assumes that the source and target share the same architecture, differing only in pre-training. This limits real-world applicability, as many practical scenarios involve transferring between models with different structures or sizes.

**Questions:**

1. Why can’t we directly use the source model? If the source model is indeed inaccessible, obtaining its task vector still seems almost equivalent to having it, since it only differs from the base pre-trained weights—usually publicly available.

2. Could the authors provide a more convincing motivation or real-world use case where such task-vector transport is necessary or clearly beneficial?

3. Is the task-vector transport still effective in multi-task scenarios—for example, when merging multiple transported task vectors?

I would be happy to raise my score if the authors can address all these concerns convincingly.

---

> ### Author Response · Authors · 2025-11-20
>
> ## On the Motivation and Practicality
> >The motivation of the setting is not entirely convincing. If the source model has already been fine-tuned on the task, it is unclear why one would not simply use that model directly. Since obtaining the task vector requires access to the fine-tuned source, the need to reapply it to a target model feels less practical, especially when the transferred performance does not surpass the source.
>
> >Could the authors provide a more convincing motivation or real-world use case where such task-vector transport is necessary or clearly beneficial?
>
> We agree that this point needs to be clarified for the practical scenario we are addressing.
>
> The motivation for our work is not a setting where the user loses their original fine-tuned model, but the core problem is model maintenance and update.
>
> Consider this common scenario:
> 1.  A user fine-tunes a task vector $\tau_A$ on a base model $A$ (e.g., a `ViT-B-16` released in 2022).
> 2.  Six months later, the model provider (e.g., Google, OpenAI) releases a new, *superior* base model $B$ (e.g., a `ViT-B-16` released in 2023) of the same architecture, but pre-trained on a much larger or cleaner dataset.
> 3.  The user's original fine-tuned model ($\theta_A + \tau_A$) is now built on an obsolete foundation. To get the performance benefits of the new base model $B$, the only option of the user is to discard $\tau_A$ and re-run the entire fine-tuning process on $\theta_B$. This is expensive, time-consuming, or sometimes impossible if the whole original (and often proprietary) fine-tuning data is no longer available.
>
> Our work, GradFix, provides a "plug-and-play" alternative. It allows the user to transport the task-specific knowledge from $\tau_A$ directly onto the new base model $\theta_B$ with minimal access to task data, often as few as $10–20$ labeled examples, without the need for a full fine-tuning process. This approach mitigates the cost of re-training from scratch and is practical even when the full original dataset is no longer accessible. The goal is to provide a "fine-tuning-free" model on a new base ($\theta_B$) that significantly outperforms naive baselines (like zero-shot $\theta_B$ or naively adding $\tau_A$).
> We believe this is a practical and valuable problem for the community, as it saves significant computational costs associated with the rapid release of new foundation models.
>
> ## Same-Architecture Assumption
> >The method assumes that the source and target share the same architecture, differing only in pre-training. This limits real-world applicability, as many practical scenarios involve transferring between models with different structures or sizes.
>
> This was a deliberate methodological choice to isolate the core problem. The challenge of transferring task vectors can be split into two distinct problems:
>
> 1. **Architectural Alignment:** Mapping parameters from model $A$ to model $B$ when their architectures differ (e.g., different layer widths, increased model size, or structural modifications).
> 2. **Knowledge Transfer:** Adapting the source task vector $\tau_A$ to the target model $\theta_B$ to leverage the pre-trained knowledge effectively.
>
> Addressing the first problem (architectural alignment) is a highly complex research area in its own right, involving challenges like solving linear assignment problems for weight permutations or finding optimal "model-stitching" solutions.
>
> Our paper focuses and isolates the second problem, which is already non-trivial. We simplified the setting to the "same-architecture" case, which, as described in our motivation, is an extremely common scenario (e.g., `T5-v1.0` -> `T5-v1.1`, or `CLIP-ViT-B-16` -> `OpenCLIP-ViT-B-16`).
>
> We fully agree that extending this work to handle different structures or sizes is a critical next step and we aim to tackle it in future works.

---

> ### Author Response · Authors · 2025-11-20
>
> ## Multi-Task Scenarios and Merging
> >Is the task-vector transport still effective in multi-task scenarios—for example, when merging multiple transported task vectors?
>
> We thank the reviewer for the insightful question. Our setting indeed differs substantially from the classical model-merging scenario: we transport task vectors across different pre-trains, where parameter spaces are not aligned. In contrast, standard merging methods (e.g., TIES, Task Arithmetic) assume that all task vectors originate from the same base initialization, which ensures direct comparability of magnitudes and signs. This assumption is precisely what our transport setting relaxes.
>
> However, we agree that the suggested multi-task transport setting is and interesting and worth to explore question. Therefore, we added new experiments in Section 5.2 and Table 3, reporting the comparison of two strategies we devised for merging multiple transported task vectors using GradFix in combination with traditional model merging methods such as Task Arithmetic and TIES-Merging:
>
> 1.  **Mask–then–Merge (Merging Transported Vectors):** We first transport each task vector individually using GradFix and then merge the resulting updates ($\theta_B - \text{Merge}(\{\delta^A_i\})$).
> 2.  **Merge–then–Mask (Transporting Merged Vectors):** We first merge the raw source task vectors (using TIES or Task Arithmetic) and then transport the merged vector using a consensus gradient mask on the target model ($\theta_B - \alpha(m_{\text{cons}} \odot \tau_{\text{merged}})$). The consensus mask $m_{\text{cons}}$ is computed in two steps: (1) a gradient-sign mask is estimated for each target dataset on $\theta_B$; (2) for each parameter, we select the sign that appears most frequently across these masks. This yields a direction that is globally descent-aligned for Model B and is used to transport $\tau_{\text{merged}}$ effectively.
>
> |Method|EURO|SVHN|GTSRB|RESISC|DTD|AVG|
> |:---|:--:|:--:|:--:|:--:|:--:|:--:|
> |$\theta_B$ zero-shot|49.41|50.58|48.29|67.98|55.96|54.44|
> |$\theta_B+\tau_A$|49.58|50.84|49.31|67.87|56.27|54.77|
> |$\theta_B−\delta_A$|65.07|70.19|**64.33**|71.42|**58.51**|65.90|
> |**TA merge**|||||||
> |TA no transport|49.31|50.99|48.73|68.05|56.54|54.73|
> |Mask→Merge|55.90|71.56|59.65|**71.40**|57.66|63.23|
> |Merge→Mask|65.37|72.10|59.55|71.16|57.07|65.05|
> |**TIES merge**|||||||
> |TIES no transport|49.15|50.75|48.95|67.97|56.54|54.67|
> |Mask→Merge|50.41|64.28|54.05|69.51|57.13|59.08|
> |Merge→Mask|**65.62**|**72.42**|62.73|**71.57**|57.77|**66.02**|
>
> These experiments show that task-vector transport and task-vector merging operate independently: transport does not interfere with merging, and merging does not disrupt transport. In practice, the two steps can be composed, allowing us to construct a reliable multi-task vector by first merging the source task vectors and then transporting the result to the target model.

---

> ### Comment · Reviewer_PYpn · 2025-11-25
>
> Thank you for your response, and I’m convinced by the performance on the multi-task setting. However, I still have a few questions:
> (1) Given that the transferred model doesn’t outperform the fine-tuned source model, why would users choose to transfer to the newer base model instead of sticking with the fine-tuned version? What’s the benefit if the performance doesn’t surpass the old one?
> (2) Does merging the transported task vector affect the model’s generalization, e.g., on held-out datasets? If the new model doesn’t surpass the old one on the target task, its real-world benefit might lie in leveraging the new model's generalization ability.

---

> > ### Author Response · Authors · 2025-11-25
> >
> > **Response to Reviewer**
> >
> > > Given that the transferred model doesn’t outperform the fine-tuned source model, why would users choose to transfer to the newer base model instead of sticking with the fine-tuned version?
> >
> > The decision to migrate to a new foundation model release $\theta_B$ is motivated not just by performance on a single task, but by the opportunity to leverage the updated backbone’s superior zero-shot capabilities. While GradFix does not fully match $\theta_A^{ft}$ on the target task, transferring to the newer base allows us to benefit from $\theta_B$’s improved zero-shot generalization. We will clarify this motivation more explicitly in the final manuscript.
> >
> > > Does merging the transported task vector affect the model’s generalization, e.g., on held-out datasets? If the new model doesn’t surpass the old one on the target task, its real-world benefit might lie in leveraging the new model's generalization ability.
> >
> > As suggested by the reviewer, we conducted a new experiment evaluating zero-shot capability on an held-out dataset (ImageNet-R) across our benchmarks. This tests whether GradFix damages the broad generalization that motivates upgrading to $\theta_B$ in the first place.
> >
> > The results (now added to Appendix C.4), confirm that GradFix-transferred model largely preserves $\theta_B$’s zero-shot performance on ImageNet-R while substantially improving target-task accuracy (e.g., +18% EuroSAT, +19% SVHN, +17% GTSRB). Although it does not fully match $\theta_A^{ft}$, the transferred model achieves a balance between broad generalization and task-specific performance, making it valuable for real-world scenarios.
> >
> > | **Method** | **EuroSAT Task** | **EuroSAT Supp.** | **SVHN Task** | **SVHN Supp.** | **GTSRB Task** | **GTSRB Supp.** | **RESISC45 Task** | **RESISC45 Supp.** | **DTD Task** | **DTD Supp.** |
> > |------------|------------------|--------------------|---------------|----------------|----------------|------------------|--------------------|----------------------|--------------|----------------|
> > | $\theta_A$ *zero-shot* | 49.08 | 68.80 | 47.00 | 68.80 | 43.37 | 68.80 | 58.94 | 68.80 | 47.50 | 68.80 |
> > | $\theta_B$ *zero-shot* | 49.41 | 79.78 | 50.58 | 79.78 | 48.29 | 79.78 | 67.98 | 79.78 | 55.96 | 79.78 |
> > | $\theta_A$ *ft* | 98.58 | 65.93 | 93.59 | 67.77 | 98.23 | 65.98 | 92.43 | 65.35 | 79.04 | 65.02 |
> > | $\theta_B − δ^A$ | 67.80 | 79.77 | 69.74 | 79.73 | 65.23 | 80.17 | 72.14 | 79.88 | 59.63 | 79.27 |

---

> ### Comment · Reviewer_PYpn · 2025-11-25
>
> Thank you for the additional clarification and experiments. Considering the potential benefit of leveraging the improved generalization ability of the newer base model, I now find the setting more convincing in practical terms. The held-out evaluation on ImageNet-R is also reassuring and strengthens the motivation. For the final version, I encourage the authors to make this practical motivation more explicit, and move this generalization table into the main text to better highlight the real-world value of this setting.
>
> Overall, I believe the paper addresses a meaningful problem, and the proposed solution has the potential to inspire follow-up work in this direction. I am raising my score to 8.

---

### Official Review · Reviewer_1QUv · 2025-11-10

**Soundness:** 3
**Presentation:** 3
**Contribution:** 3
**Rating:** 6
**Confidence:** 4

**Summary:**

The paper proposes GradFix, a method to transfer task-specific fine-tuning knowledge from a source pre-trained model to a different target pre-trained model. The authors argue that naive transfer fails due to parameter space misalignment and that the key to successful transfer lies in the sign structure of the target model's gradients. GradFix computes the gradient on the target model using only a few labeled samples. It then creates a binary mask by checking for sign agreement between the source task vector and the target gradient. This mask filters the source task vector and updates the target model parameters without any fine-tuning. The authors provide a theoretical guarantee that this update ensures a first order loss descent on the target model. Empirically, GradFix is shown to outperform naive task vector addition and standard few-shot fine-tuning on vision and language benchmarks.

**Strengths:**

1. The challenge of updating downstream tasks when a new foundation model is released is a significant and practical problem for the ICLR community.
2. The core idea of using the target model's gradient signs as a filter for the source task vector is elegant and easy to understand. The method's simplicity of requiring only a single forward-backward pass on a few labeled samples is a major practical advantage.
3. Figure 1 provides an excellent and unambiguous illustration of the core masking procedure.
4. While simple, the proof of a first-order descent guarantee (Sec 4.2)  provides a solid theoretical justification for why the method should work, moving it beyond a simple heuristic.
5. The use of majority voting to estimate gradient signs from a handful of samples (Sec 4.3) is well-justified, and the accompanying concentration analysis (Appendix A)  strongly supports this design choice.
6. The evaluation spans both vision (CLIP ViT) and language (T5) models, demonstrating the applicability of the approach across domains.

**Weaknesses:**

# Major Concerns

1. **Over-reliance on a weak baseline $\theta_B^{opt}$**: The primary few-shot baseline is defined as fine-tuning the target model with the same few samples $\mathcal{D}_s$. However, Appendix B.2 clarifies that this is achieved with a "single step of gradient descent". This is not a standard definition of few-shot fine-tuning. A more conventional baseline would involve multi-epoch training on $\mathcal{D}_s$ or applying a parameter-efficient method like LoRA on $\mathcal{D}_s$. As such, the claims of consistently outperforming few-shot finetuning are based on a potentially weak comparison, which may inflate GradFix's perceived gains.

2. **Underspecified hyperparameter $\alpha$**: The method introduces a critical scaling factor $\alpha$ in Equation 5, whjich is tuned on a validation set for each dataset (in Table 4). This undermines the "plug-and-play" claim in a strict few-shot setting where a representative validation set may not exist. The sensitivity analysis in Section 5.4 confirms that performance is sensitive to $\alpha$, but it does not propose a way to set $\alpha$ _without_ a validation set. This is a key practical limitation.

3. **Complete disregard for gradient magnitude**: The method relies _only_ on sign agreement, discarding all magnitude information from both the source task vector $\tau_A$ and the target gradient $g_B$. This seems like an extreme simplification. A component of $\tau_A$ with a very large, task-critical magnitude will be zeroed out if it has a sign mismatch. Conversely, a tiny-magnitude component of $\tau_A$ is preserved as long as its sign matches. The justification for this sign-only choice is not deeply explored.

## Minor Concerns

1. In Table 1, the full $\theta_B \textit{fine-tune}$ performance is slightly higher than $\theta_B + \delta^*$ (oracle) performance (e.g., EUROSAT: 98.70 vs 95.06; GTSRB: 98.65 vs 82.92). This suggests that $\tau_A$ is fundamentally missing significant information (or contains conflicting information) that cannot be recovered even by a perfect sign mask based on $\tau_B$. This information gap is a limitation and needs more discussion.

2. In Section 5.2, the paper compares "agreement" with "force agreement". The argument is that "agreement" is more robust to noisy gradients. However, "force agreement" seems more conceptually aligned with the theoretical goal of ensuring a descent direction $g^\top \delta^A \geq 0$. This choice feels non-obvious and the empirical difference (Table 3) is small, making the justification feel post-hoc.

3. he finding that random sampling is a strong baseline compared to structured methods like coreset or herding is a good practical result . However, it is also surprising. This might imply that the gradient sign structure is extremely stable across the data distribution, which is an interesting finding in itself and could be analyzed more deeply.

**Questions:**

1. Could the authors please justify the "single step of gradient descent"  for the $\theta_B^{opt}$ baseline? How does GradFix compare to a more standard few-shot baseline, such as training $\theta_B$ on $\mathcal{D}_s$ for maybe 10 epochs, or training a PEFT method on $\mathcal{D}_s$?

2. In a practical, no-validation-set scenario, how should $\alpha$ be set? What is the performance of GradFix if $\alpha$ is fixed to a default value (e.g., $\alpha = 0.5$) across all datasets in Table 1?

3. Have you explored hybrid approaches that do not discard magnitude entirely? For instance, what happens if you down-weight sign-mismatched components instead of zeroing them, or use a method that combines signs with magnitudes (e.g, TIES-Merging style?)

4. The related work on TIES-Merging  also focuses on resolving sign conflicts for task vectors. How does GradFix's "masking" approach conceptually differ from TIES's "sign-consistency" aggregation? How would GradFix compare empirically if TIES was adapted for this transport setting?

---

> ### Author Response · Authors · 2025-11-20
>
> ## Justification for single step gradient descent
> >Could the authors please justify the "single step of gradient descent" for the $θ_B^{opt}$ baseline?
>
> Our method, is designed as a "transport" mechanism rather than a training procedure. It requires only a single forward-backward pass to estimate gradient signs, followed by a computationally negligible element-wise vector operation. To isolate the specific contribution of the **transported task vector** (as opposed to information gained purely from the new data), we compared GradFix against a baseline constrained to the **same computational budget**. Specifically, the baseline must use:
>
> * the **same data** $\mathcal{D}_s$,
> * the **same gradient** at $θ_B$, and
> * the **same number of forward–backward passes**.
>
> Under these constraints, the natural competitor is $θ_B^{opt}$, the single-step optimizer update using the same gradient on the same data. This ensures that both methods rely on identical gradient information and incur the same forward–backward cost; only the post-gradient update rule differs.
> Thus, $θ_B^{opt}$ is the compute-matched counterpart that lets us determine whether the **masking itself** provides value beyond a single direct gradient update.
>
> >How does GradFix compare to a more standard few-shot baseline?
>
> In contrast, standard few-shot fine-tuning or LoRA involve many optimizer steps and consequently require many forward–backward passes.
>
> To clarify this distinction, we included a detailed computational cost analysis in **Appendix F**, showing that multi-epoch fine-tuning operates at a different computational scale and is therefore not comparable to our method. GradFix is inherently a single-step transport procedure requiring only $8P$ FLOPs for $P$ total parameters, comparable to a single optimization step $\theta_{B}^{opt}$ at $16P$ FLOPs and roughly $4000$ times more efficient than the full $2000$-step fine-tuning used to obtain the source task vectors. Our comparison to $\theta_{B}^{opt}$ is thus intended to validate GradFix against a single-step baseline under the same compute and data constraints, demonstrating that our selective transport mechanism yields a more effective update than simple local optimization.
>
> ## Alpha tuning
> > The method introduces a critical scaling factor $\alpha$ in Equation 5… This undermines the "plug-and-play" claim… The sensitivity analysis in Section 5.4 confirms that performance is sensitive to $\alpha$… This is a key practical limitation.
>
> Determining the optimal magnitude for parameter updates remains an open research challenge across the task vector and model merging literature; methods such as Task Arithmetic (Ilharco et al., 2023), TIES-Merging (Yadav et al., 2023), KnOTS (Stoica et al., 2024), TSV (Gargiulo et al., 2025), Iso-C (Marczak et al., 2025), MagMax (Marczak et al., 2024), AdaMerging (Yang et al., 2024), and DARE (Yu et al., 2024) determine this value via a hyperparameter sweep on held-out data.
>
> Our work inherits this characteristic; however, our specific masking mechanism offers distinct stability advantages. As detailed in **Section 5.4** and **Figure 2**, our sensitivity analysis demonstrates that the **Majority Voting** mask strategy acts as a natural regularizer, yielding smooth performance curves that maintain high accuracy across a broad range of $\alpha$ values. This inherent stability implies that while tuning $\alpha$ yields the theoretical optimum, precise tuning is not strictly necessary to achieve significant improvements while using GradFix. In fact, for most datasets, its performance is stable within the range $\alpha \in [0.3, 0.7]$.
> > In a practical, no-validation-set scenario, how should  be set? What is the performance of GradFix if  is fixed to a default value (e.g., 0.5 ) across all datasets in Table 1?
>
> Consequently, for scenarios without a validation set, as pointed out by the reviewer, we suggest setting a fixed **$\alpha = 0.5$** as a robust default. To demonstrate the viability of this heuristic, we report the performance of GradFix with this fixed $\alpha = 0.5$ and compare it to the Naive Transfer applied with the same fixed $\alpha$.
>
> **ViT-B/16 Results**
> | Dataset | Naive Transfer ($\theta_B + \tau_A$) | GradFix | Gain |
> | :--- | :--- | :--- | :--- |
> | **EuroSAT** | 49.14% | 60.63% | +11.49% |
> | **SVHN** | 50.62% | 65.37% | +14.75% |
> | **GTSRB** | 55.12% | 61.37% | +6.25% |
> | **RESISC45**| 65.57% | 68.33% | +2.76% |
> | **DTD** | 56.10% | 58.40% | +2.30% |
>
> **ViT-L/14 Results**
> | Dataset | Naive Transfer ($\theta_B + \tau_A$) | GradFix | Gain |
> | :--- | :--- | :--- | :--- |
> | **EuroSAT** | 62.33% | 69.50% | +7.17% |
> | **SVHN** | 38.82% | 73.26% | +34.44% |
> | **GTSRB** | 55.75% | 66.99% | +11.24% |
> | **RESISC45**| 72.43% | 77.80% | +5.37% |
> | **DTD** | 62.51% | 65.21% | +2.70% |

---

> ### Author Response · Authors · 2025-11-20
>
> ## On the Magnitude
> > Complete disregard for gradient magnitude: The method relies only on sign agreement… A component of $\tau_A$ with a very large, task-critical magnitude will be zeroed out if it has a sign mismatch… Conversely, a tiny-magnitude component is preserved if its sign matches… The justification for this sign-only choice is not deeply explored.
>
> > Have you explored hybrid approaches that do not discard magnitude entirely?
>
> We agree with the reviewer about the underexploration of the effect of the magnitude of the masking strategy. We want to highlight that the mean strategy, reported in **Fig. 2**, already takes into account the magnitude of the gradients and leads to consistently worse and less robust results.
>
> Additionally, during our preliminary experiments, we also evaluated other magnitude-based approaches. Specifically, we evaluated a masking strategy named **weighted agreement**. This method incorporates magnitude information from *both* the source task vector ($t$) and the target gradient ($g$). The mask $M$ for this strategy is computed as:
>
> $M_j = \max(0, \tanh((t_j \cdot g_j)))$
>
> This mask is directly proportional to the magnitudes of both vectors. A large, matching component in $t$ and $g$ results in a strong positive mask (close to $1$), while a mismatch in sign *or* a tiny magnitude in either component results in a mask of $0$. This directly tests whether the magnitude information discarded by our original method is valuable or not.
> We have incorporated this strategy into the main body of the revised paper and provide a discussion in **Section 5.3**, with results summarized in the masking strategies **Table 5**. Our findings show that magnitude-aware weighted agreement consistently underperforms the sign-only agreement mode.
> This result provides empirical evidence that, while counterintuitive, the magnitude of the approximated gradient appears to be a noisy signal. We sincerely thank the reviewer, as their suggestion has allowed us to strengthen the justification for our method.

---

> ### Author Response · Authors · 2025-11-20
>
> ## Relation between TIES-Merging and GradFix
>
> > The related work on TIES-Merging focuses on resolving sign conflicts… How does GradFix's "masking" approach differ from TIES's "sign-consistency" aggregation?… How would GradFix compare empirically if TIES were adapted for this transport setting?
>
> While both TIES-Merging and GradFix utilize sign information to resolve conflicts, the two approaches address different problems, and are **complementary** rather than competing.
>
> **TIES-Merging** is designed for combining multiple task vectors that all originate from the same base model. Under this assumption, parameter magnitudes are directly comparable and magnitude pruning plus sign consistency can effectively suppress harmful interference.
>
> In contrast, **GradFix** tackles a different challenge: transferring a single task vector across different pre-trains (_i.e._, $\theta_A \leftarrow \theta_B$), where parameter spaces are *not aligned*, and magnitudes of $\tau_A$ have no reason to be meaningful for $\theta_B$. In this cross-pretrain setting, magnitude-based heuristics become unreliable, whereas the gradient sign of the target model offers a model-specific and geometry-aware signal that indicates locally beneficial directions. This motivates the gradient-aligned masking strategy at the core of GradFix.
>
> To empirically investigate the interaction between task vector merging and transport, we conducted experiments on ViT-B/16, combining GradFix two popular merging methods, TIES and Task Arithmetic. These results are presented in the updated **Section 5.2** and **Table 3** of the revised paper. Specifically, we evaluate two integration strategies:
>
> * **Mask–then–Merge** ($\theta_B - \text{Merge}(\{\delta^A_i\})$). We first transport each individual task vector using GradFix (masking via target gradients), and then merge the resulting updates.
> * **Merge–then–Mask** ($\theta_B - \alpha(m_{\text{cons}} \odot \tau_{\text{merged}})$) resolves inter-task conflicts before transport. First, TIES (or Task Arithmetic) merges the raw source task vectors into a unified task vector ($\tau_{\text{merged}}$). Next, the consensus gradient mask ($m_{\text{cons}}$) for $\theta_B$ is computed in two steps: (1) a preliminary gradient sign mask is computed for each target dataset; (2) for each parameter, $m_{\text{cons}}$ adopts the sign appearing most frequently across these masks. This mask captures the globally descent-aligned direction on $B$, which is then applied to transport $\tau_{\text{merged}}$.
>
> |Method|EURO|SVHN|GTSRB|RESISC|DTD|AVG|
> |:---|:--:|:--:|:--:|:--:|:--:|:--:|
> |$\theta_B$ zero-shot|49.41|50.58|48.29|67.98|55.96|54.44|
> |$\theta_B+\tau_A$|49.58|50.84|49.31|67.87|56.27|54.77|
> |$\theta_B−\delta_A$|65.07|70.19|**64.33**|71.42|**58.51**|65.90|
> |TA merge|||||||
> |TA no transport|49.31|50.99|48.73|68.05|56.54|54.73|
> |Mask→Merge|55.90|71.56|59.65|**71.40**|57.66|63.23|
> |Merge→Mask|65.37|72.10|59.55|71.16|57.07|65.05|
> |TIES merge|||||||
> |TIES no transport|49.15|50.75|48.95|67.97|56.54|54.67|
> |Mask→Merge|50.41|64.28|54.05|69.51|57.13|59.08|
> |Merge→Mask|**65.62**|**72.42**|62.73|**71.57**|57.77|**66.02**|
>
> The results in the table show a clear pattern:
>
> * **Mask–then–Merge** performs poorly with TIES. Applying masks individually introduces sparse and heterogeneous patterns that disrupt the parameter structure TIES relies on.
> * **Merge–then–Mask** consistently performs the best, often matching or exceeding single-task GradFix. TIES first resolves inter-task conflicts in the source domain, producing a clean merged vector, then GradFix aligns this unified update to the geometry of $\theta_B$.
>
> Importantly, even when TIES produces a high-quality merged vector, GradFix remains essential to make that vector transferable across pretrains.
>
> Additionally, we notice that the merging operations further alleviates the sensitivity to the selection of $\alpha$, especially under the Merge-than-Mask strategy. Indeed, TIES first removes destructive magnitude interference across tasks, producing a balanced and stable merged vector. Then, GradFix applies a single global mask aligned with the target geometry, suppressing spurious coordinates from $\tau_A$. As a result, the merged-and-masked vector behaves much more consistently across datasets, and a single fixed $\alpha = 1$ works remarkably well without tuning.
>
> To sum up, we have the following findings:
> * Transport and merging are **orthogonal operations**. Merging determines how multiple tasks interacts within the source space, while GradFix determines whether the resulting update is meaningful for the target model.
> * Merge-then-Mask yields a robust multi-task transported vector with a reduced sensitivity to $\alpha$ selection.
> * TIES benefits more than Task Arithmetic from GradFix transport, confirming that GradFix provides information that magnitude pruning alone cannot recover across pretrains.

---

> ### Author Response · Authors · 2025-11-20
> **Minor Concerns**
>
> ### Information Gap (Oracle vs. Full Fine-Tuning)
> >In Table 1, the full $\theta_B\ fine-tune$ performance is slightly higher than $\theta_B + \delta^\star$ (oracle) performance (e.g., EUROSAT: 98.70 vs 95.06; GTSRB: 98.65 vs 82.92). This suggests that $\tau_A$  is fundamentally missing significant information (or contains conflicting information) that cannot be recovered even by a perfect sign mask based on $\tau_B$. This information gap is a limitation and needs more discussion.
> The reviewer correctly observes that the gap between the full fine-tune ($\theta_{B}^{ft}$) and the oracle transfer ($\theta_{B}+\delta^{*}$) shows that the source task vector $\tau_A$ lacks information needed for optimal adaptation of Model B.
>
> The gap in performance reflects a core difference between **transporting** an existing solution and **optimizing** a new one:
>
> * $\tau_A$: Derived from a different pre-training ($\theta_A$), encoding how Model A adapts to the task.
> * $\tau_B$: The true task vector for Model B emerges from a full optimization trajectory starting at $\theta_B$, introducing information, such as corrections to $\theta_B$-specific feature biases, that is absent from $\tau_A$.
>
> GradFix transports the information available in $\tau_A$ to the model $B$ by filtering it through the loss landscape of the model $B$ itself, but it cannot replace the model-specific information that would be produced during full fine-tuning. The performance gap (e.g., GTSRB: 82.92% for $\theta_{B}+\delta^{*}$ vs. 98.65% for $\theta_{B}^{ft}$) quantifies how much of the final solution is unique to Model B’s own optimization path.
>
> ### Agreement vs. Force Agreement
> > In Section 5.2, the paper compares agreement with force agreement. The argument is that agreement is more robust to noisy gradients. However, force agreement seems more conceptually aligned with the theoretical goal of ensuring a descent direction $g^\top \delta^A \ge 0$ . This choice feels non-obvious and the empirical difference (Table 3) is small, making the justification feel post-hoc.
>
> We appreciate the opportunity to clarify the distinction between “Agreement” and “Force Agreement.”
> Although their empirical performance differs only marginally in the few-shot regime (e.g., ~0.27% on average in Table 5 (old Table 3) ), we argue that **Agreement** is a more robust choice when gradients are noisy. The key issue is the asymmetry of risk when the gradient estimate $\hat{s}$ is incorrect:
>
> * **With the Oracle:** Using the true task vector $\tau_B$, “Force Agreement” performs best, precisely because the direction is exact.
> * **With Estimated Gradients:** When gradients are few-shot (sometimes from a single sample), “Agreement” typically outperforms “Force Agreement.”
>
> While average performance is similar, Agreement avoids committing to harmful directions when gradient estimates are unreliable. In such cases, it is generally safer to omit a potentially useful update than to enforce a wrong direction driven by a weak or noisy estimate.

---

> > ### Comment · Reviewer_1QUv · 2025-11-27
> >
> > Thank you for the detailed clarifications! I understand the distinction between **transport** and o**optimization** now. I believe this paper is strong and should be accepted. I am increasing my score to 8, in hope that the authors will commit to proving the results that are still running in the camera ready version.

---

### Author Response · Authors · 2025-11-21
**Summary of Changes**

We sincerely thank all reviewers for their constructive and insightful feedback. Their comments helped us significantly improve the paper. In particular, we revised the manuscript and made the following additions:

* **Weighted sign-agreement metric** (Table 5):
We introduced a new analysis quantifying the agreement between the source task vector and the target gradient field, clarifying why masking works.

* **Multi-source / multi-task experiments** (Section 5.2):
New experiments combining GradFix with Task Arithmetic and TIES demonstrate that merging and transport are complementary and can be composed effectively.

* **Random-vector ablation** (Appendix C.3):
We added an ablation replacing the real task vector with a random vector to verify that GradFix’s gains do not arise from trivial sparsification or noise filtering.

* **Computational comparison** (Appendix F):
We added a runtime and compute analysis showing that a one-step gradient update is the correct baseline relative to GradFix.

* **Additional clarifications** (in the responses):
We provided further explanations regarding also the relation to TIES, $\alpha$-sensitivity, cross-scale applicability, and extensions to diffusion and VLM models.

We thank the reviewers again for their thoughtful comments and suggestions, which substantially improved the clarity and rigor of the paper.

---

### Author Response · Authors · 2025-12-02
**Summary of Rebuttal**

We provide a concise overview of our rebuttal and sincerely thank the new AC for their effort in this challenging situation.

Two reviewers (**1QUv**, **PYpn**) raised their scores to **8**. One reviewer (**sW3e**) raised their score to **6**. Two reviewers (**G6x6**, **VKR3**) have not responded to the rebuttal.

---

### Reviewer 1QUv
**(Score raised to 8)**: The reviewer finds the paper strong, now clearly understands the distinction between transport and optimization, and supports acceptance by raising their score to 8.

* **[Justification for the single-step gradient descent baseline ($\theta_{B}^{opt}$)]**: Clarified that $\theta_{B}^{opt}$ is the correct compute-matched baseline for our method and added a computational cost analysis (App. F).
* **[A practical strategy for setting the hyperparameter $\alpha$ without a validation set]**: Demonstrated that a fixed default $\alpha=0.5$ yields robust performance across datasets, outperforming naive transfer.
* **[Investigation into why gradient magnitude is discarded in mask creation and if hybrid approaches (magnitude + sign) would work better]**: Implemented a "weighted agreement" strategy (magnitude-aware), showing it consistently underperforms the proposed sign-only masking, proving magnitude is a noisy signal in this setting (Sec. 5.3 & Tab. 5).
* **[Comparison/Relation to TIES-Merging]**: Added experiments combining GradFix with TIES and Task Arithmetic, showing "Merge-then-Mask" is effective and complementary (Sec. 5.2 & Tab. 3).

---

### Reviewer PYpn
**(Score raised to 8)**: The reviewer finds the paper practically convincing, appreciates the additional experiments and generalization evaluation, sees potential for impact, and raises their score to 8.


* **[Why transport a task vector if the source fine-tuned model already exists?]**: Clarified the use case: upgrading an old fine-tuned model to a new foundation model release to leverage improved zero-shot capabilities without full re-training (App. C.4).
* **[Validation of effectiveness in multi-task scenarios when merging multiple transported vectors]**: Conducted multi-task experiments showing GradFix can successfully transport merged vectors (Sec. 5.2 & Tab. 3).
* **[Clarification on the "same-architecture" limitation]**: Clarified that we deliberately separate knowledge transfer from the distinct and complex challenge of architectural alignment.
* **[(In discussion) Evidence that the transported model generalizes to held-out datasets]**: Added ImageNet-R evaluation, proving GradFix preserves the new base model's zero-shot generalization better than the old fine-tuned source (App. C.4).

---

### Reviewer G6x6
**(Reviewer has not responded)**

* **[Applicability to cross-scale models (different sizes)]**: Explained that cross-scale transport requires a parameter mapping function (an open research area), but GradFix is compatible once that mapping exists.
* **[Evaluation on multi-source transfer (multiple source models to one target)]**: Added a multi-source experiment, showing "Mask-then-Merge" outperforms single-source transport (Sec. 5.2 & Tab. 4).
* **[Applicability to Diffusion models or VLMs]**: Clarified theoretical applicability to any differentiable loss (including Diffusion/VLMs).

---

### Reviewer sW3e
**(Score updated to 6)**: The reviewer appreciates the clarifications and ablation, acknowledges the method’s effectiveness in low-data settings, and accordingly updated their score to 6.

* **[Ablation study to verify improvements are due to task knowledge transfer and not just exploiting a random vector via gradient alignment]**: Performed a random-vector ablation (Gaussian vector), showing it performs significantly worse than GradFix, confirming valid knowledge transfer (App. C.3 & Tab. 8).
* **[Discussion on the performance gap between GradFix and full fine-tuning, and the diminishing returns of adding more samples]**: Explained the gap: GradFix aligns an initial descent direction, which is fundamentally different from a converged multi-epoch solution. The reviewer acknowledged this and validated the effectiveness for the low-data regime.

---

### Reviewer VKR3
**(Reviewer has not responded)**

* **[Clarification on the limitation of the same-architecture assumption]**: Acknowledged the same-architecture constraint as a deliberate scope to focus on knowledge transfer, leaving architectural alignment for future work.
* **[Strategy for $\alpha$ selection given the sensitivity shown in figures]**: Proposed the fixed $\alpha=0.5$ strategy which works robustly without tuning.
* **[Analysis of performance variance and the "crossover point" where standard fine-tuning beats GradFix]**: Provided reference to variance tables (App. B.1 & Tab. 6 & Tab. 7) showing GradFix has lower variance than the baseline. Specified the crossover point: GradFix is superior for ViT-B/16 up to 50 shots (68.15% vs. 66.26%) and for ViT-L/14 up to 10–20 shots (10: 73.77% vs. 70.46%; 20: 74.15% vs. 74.31%).

---

### Meta-Review · Area_Chair_dKu4 · 2026-01-07

**Summary:**

This paper tackles a practical maintenance problem: reusing an existing fine-tuning “task vector” when a new pretrained checkpoint is released, without re-running full fine-tuning. The proposed GradFix (gradient-sign masking) is simple, compute-light (single forward-backward on a few labeled samples), and comes with a first-order descent guarantee. After rebuttal, the key concerns about baseline fairness, alpha selection without validation, whether “is it just a random vector,” and multi-task/multi-source behavior were substantially addressed with new experiments and clearer framing. Remaining limitations are mainly scope (same-architecture) and the expected gap to full fine-tuning. Overall, the submission is above the acceptance bar as a clean, useful contribution in my opinion.

**Reviewer Concerns:**

**Addressed in rebuttal**

* Baseline fairness / compute-matching: clarified that the intended comparison is to a single-step gradient update under the same forward-backward budget; added compute analysis.
* Hyperparameter alpha without validation: provided a robust fixed default alpha with strong gains over naive transfer across datasets.
“Maybe it works with any vector” concern: added random-vector ablation showing materially worse performance than transporting real task vectors.
* Motivation / why transport if source exists: clarified the upgrade setting (new base model has better zero-shot generalization), plus added held-out generalization evidence (ImageNet-R) supporting the practical value.
* Compositionality: added multi-task merge and multi-source transfer experiments showing transport composes with merging and can benefit from multiple sources.
* Magnitude vs sign: tested magnitude-aware variants and found them less robust than sign-only masking.

**Remaining gaps:**

* Same-architecture requirement: still a real limitation; cross-scale or architectural changes require an external parameter mapping, which is out of scope here in my opinion, and can be studied in future works.
* Gap to full fine-tuning: remains substantial on some tasks and cannot be closed by more samples in the same way as iterative fine-tuning; this should be emphasized as a trade-off (single-step transport vs optimization).
* External validity of the “few-shot fine-tuning” claim: the paper now clarifies the compute regime, but readers may still want clearer messaging that this is not a replacement for multi-epoch PEFT in higher-data settings. This can be fixed in the camera ready version.

**Reviewer Scores:**

Reviewer 1QUv: updated to 8 (already participated; no further change expected).

Reviewer PYpn: updated to 8 (already participated; no further change expected).

Reviewer sW3e: updated to 6 (already participated; no further change expected).

Reviewer G6x6: score was already 6, since multi-source experiments and broader applicability clarifications address their main questions, with the same-arch limitation still keeping it from being a strong accept.
Reviewer VKR3: score was already 6, given added variance tables, default alpha evidence, and clarified crossover points; same-arch and residual sensitivity keep it from a clear 8.

---

### Decision · Program_Chairs · 2026-01-26

Accept (Poster)